# Infection rate models for COVID-19: Model risk and public health news sentiment exposure adjustments

**Ioannis Chalkiadakis**[1]*, **Hongxuan Yan**[2,5], **Gareth W. Peters**[3], **Pavel V. Shevchenko**[4,6]

**1** Department of Computer Science, School of Mathematical and Computer Sciences, Heriot-Watt University, Edinburgh, Scotland, United Kingdom, **2** Academy of Mathematics and Systems Science, Chinese Academy of Sciences, Beijing, China, **3** Department of Actuarial Mathematics and Statistics, School of Mathematical and Computer Sciences, Heriot-Watt University, Edinburgh, Scotland, United Kingdom, **4** Department of Actuarial Studies and Business Analytics, Macquarie University, Sydney, Australia, **5** Centre for Forecasting Science, Chinese Academy of Sciences, Beijing, China, **6** Center for Econometrics and Business Analytics, St. Petersburg State University, Saint Petersburg, Russia

* ic14@hw.ac.uk

**Data Availability Statement:** Data and code are available at the following GitHub repository: https:// github.com/ichalkiad/covid19modelrisk.

## Abstract

During the COVID-19 pandemic, governments globally had to impose severe contact restriction measures and social mobility limitations in order to limit the exposure of the population to COVID-19. These public health policy decisions were informed by statistical models for infection rates in national populations. In this work, we are interested in modelling the temporal evolution of national-level infection counts for the United Kingdom (UK—Wales, England, Scotland), Germany (GM), Italy (IT), Spain (SP), Japan (JP), Australia (AU) and the United States (US). We model the national-level infection counts for the period January 2020 to January 2021, thus covering both the pre- and post-vaccine roll-out periods, in order to better understand the most reliable model structure for the COVID-19 epidemic growth curve. We achieve this by exploring a variety of stochastic population growth models and comparing their calibration, with respect to in-sample fitting and out-of-sample forecasting, both with and without exposure adjustment, to the most widely used and reported growth model, the Gompertz population model, often referred to in the public health policy discourse during the COVID-19 pandemic. Model risk as we explore it in this work manifests in the inability to adequately capture the behaviour of the disease progression growth rate curve. Therefore, our concept of model risk is formed relative to the standard reference Gompertz model used by decision-makers, and then we can characterise model risk mathematically as having two components: the dispersion of the observation distribution, and the structure of the intensity function over time for cumulative counts of new infections daily (i.e. the force of infection) attributed directly to the COVID-19 pandemic. We also explore how to incorporate in these population models the effect that governmental interventions have had on the number of infected cases. This is achieved through the development of an exposure adjustment to the force of infection comprised of a purpose-built sentiment index, which we construct from various authoritative public

**Funding:** The authors received no specific funding for this work.

**Competing interests:** The authors have declared that no competing interests exist.

health news reporting. The news reporting media we employed were the New York Times, the Guardian, the Telegraph, Reuters global blog, as well as national and international health authorities: the European Centre for Disease Prevention and Control, the United Nations Economic Commission for Europe, the United States Centres for Disease Control and Prevention, and the World Health Organisation. We find that exposure adjustments that incorporate sentiment are better able to calibrate to early stages of infection spread in all countries under study.

## 1 Introduction

At the end of 2019, a new coronavirus strain led to the onset of a global pandemic that has ravaged the world throughout 2020 and continues into 2021, termed generically the COVID-19 respiratory disease. It has had an immense impact on society in a multitude of ways, with significant mortality (according to the weekly epidemiological update of the World Health Organisation for the week of March 16 2021, there have been over 119M confirmed cases and over 2.6M deaths) and morbidity, long term health effects (long-COVID) and significant impact on global economies. It is therefore important to study retrospectively the statistical properties of the evolution of this disease to try to address statistical questions such as "Why were so many disease growth rate projections so significantly wrong in the early stages of the pandemic?". We seek a partial answer to this question from a statistical perspective based on an analysis of model risk. In addressing this question, we then gain insight on two additional questions, namely "What is the most reliable and accurate way to build an epidemic growth model for this disease?" and "Can one assess the influence of public policy and public health reporting on the dynamics of the COVID-19 pandemic spread over time?". The significance of finding statistical models to address such questions is apparent, as they can provide greater insight into numerous aspects of the pandemic.

We are also interested in obtaining a greater understanding of how the communication of public health announcements influenced the populations' behaviour and whether this had a marked effect on flattening the curve. We quantified this through changes in infection rates each day as a result of public health policy and information announcements. In order to quantify the public health information to incorporate it into our stochastic infection growth rate models as an exposure adjustment, we had to obtain a daily time-series summary of this health reporting. In this regard, we developed a novel Natural Language Processing (NLP) sentiment index that we extracted over time via text mining from a variety of press releases and news articles that we extracted from authoritative news agencies and public health authorities that included the New York Times (NYT), the Guardian, the Telegraph, Reuters global blog, the European Centre for Disease Prevention and Control (ECDC), the US Centre for Disease Control and Prevention (USCDC), the World Health Organisation (WHO) and the UN Economic Commission for Europe (UNECE).

All of these reporting sources have a global reach. In particular, the NYT attract a worldwide audience, as they provide reporting for news happening globally, either from in-house journalists, or re-publishing reporting from global news agencies (19% of the NYT-published articles we processed were in fact coming from Reuters). Furthermore, 16% of the NYT subscribers come from outside the US (source: https://letter.ly/new-york-times-readership-statistics/), excluding its non-subscribed readers, and the country with the fastest growing NYT readership is Australia, one of the countries we focus on. In addition, the Guardian offers

significant coverage for European audiences as more than 50% of its readership comes from all over Europe (source: https://www.theguardian.com/advertising/audience1). In total, we considered 37, 066 articles that had undergone editorial screening from authoritative global news sources.

There are numerous ways that one can seek to model epidemics, through for instance compartmental epidemic models, typically based on Susceptible-Exposed-Infected-Recovered models that capture individuals in compartments of stages of health, infected, recovered or deceased, and sometimes also involve more detailed compartmentalisation of the exposed population to account for other social, demographic or age-based features; numerous studies of this type have been developed for COVID-19, see examples in [1–4]. These models allow for age structure and mobility features to be incorporated and are useful for a detailed epidemiological analysis and analysis of vaccine response. The challenge with these models in the context of COVID-19 modelling is that they often rely upon a few key parameters in the calibration that strongly affect the model outputs. One of these components that is hotly debated in the epidemiological studies of COVID-19 is related to the reproductive number; see discussions on the challenge of quantifying this key component in the work of [5, 6]. Other approaches to such modelling include stochastic epidemic models, for instance, the work of [7] who focused more on continuous-time stochastic models based on a four-dimensional stochastic differential equation (s.d.e.) formulation, where the time-dependent input parameters include the reproduction number, the average number of externally new infected and the average number of new vaccinations. In order for these s.d.e. models to be feasible to use in practice, it is often required to make potentially overly simplifying assumptions to obtain tractability.

It is important to recognise that different models will be fit for different purposes, and whilst these aforementioned models are required for intricate epidemiological disease modelling at the individual demographic level and to better understand vaccine program responses, they are not the models used by public health policymakers. Often, throughout the COVID-19 pandemic's evolution, the public health decision-makers in government resorted to abstraction from such detailed models and instead focused on simpler macro infection growth rate models suitable for national-level epidemic analysis. The most popular of these models was the Gompertz model, as discussed in [8]. As in this work, we also seek to explore the models that policymakers used for public health decision making. Therefore, we also rely on national-level aggregated data since we are not aiming at modelling the micro-structure of the propagation of the disease, but instead, we seek the policymakers' perspective of the large-scale population-level predictions. Unlike in the work of [8] where the focus was on the simplest deterministic Gompertz model formulation (see Gompertz law, [9]), we have extended this model perspective significantly in four major ways: firstly, we developed stochastic latent factor GLARMA (generalised linear autoregressive moving average) count time-series models that incorporate stochastic population growth models, the simplest of which has a Gompertz structure. To this end, we consider a variety of different regression model structures on aggregate cumulative daily new infections at a national level, allowing us to analyse the trend of the propagation dynamics. Secondly, we modified the observation distribution to better accommodate an early epidemic mixing phase and a wider community mixing phase, through the use of a splice model that combined a Generalised Poisson model and a continuous observation model. Thirdly, we developed a Bayesian formulation for the estimation of these models with posterior uncertainty quantification via posterior credible intervals; and finally, we incorporated a unique exposure adjustment feature that involved

the introduction of a novel public health news announcement sentiment index. This feature allows public health officials to quantify formally the impact of public health announcements and news releases directly in terms of how they influenced the population's behaviour, as reflected by daily changes in infections as people learned more about COVID-19 through health and news releases.

By introducing sentiment information in the exposure adjustments in the models, as captured by public news, we are able to inform public health decision-makers on the effectiveness of their public information and health campaigns. Public sentiment is a way to quantify society's reaction to governmental handling of the pandemic, and whether or not people adhere to the measures taken, compulsory or not. This element, however, is inherently related to the number of infected cases, and therefore we aim to incorporate it in our models through the exposure of the observation distribution. Publicly available news and social media are a suitable proxy to capture this signal. First, note that micro-blogging (e.g. Twitter) sources have been successfully used to inform predictive models about epidemic evolution as shown in the research works of [10, 11], whereas the work of [12] analysed Twitter to detect a forthcoming outbreak. All of these studies emphasise the need to filter Twitter messages and use only those that are relevant to determine disease infection numbers—which is not trivial in general. Second, the language of news or social media posts changes over the course of an epidemic, and those changes can be informative about the epidemic and its consequences [13, 14]. In our work, we focus on more information-rich text sources based on editorial screened public health messaging, news reports and announcements. These are more reliable than basic social media information such as Twitter, but they are fewer in number. This is not a problem in the context of our application which studies daily epidemic evolution. We explain in detail how we extract the sentiment score and how we incorporate this into the study of epidemic growth models.

We note that one could also study a variety of different time-series of data: daily death counts, new infections, total cumulative infections, number of hospitalisations etc. We have selected to study, as in these aforementioned works, the daily number of infections at the national level. We are interested to model this quantity given by the number of infected, as the number of deaths is often misreported or under-reported, and is subject to reporting schedules rather than exact daily counts, due to policy and overloaded hospital and coroner reporting systems in the pandemic, hence making the use of this data less reliable in the types of models we seek to develop.

We then demonstrate how the extensions we introduce can capture model risk as it manifests in two model components: the dispersion of the observation distribution and the structure of the intensity. This could provide valuable insight both whilst this pandemic continues to rage, and as guidance to future such events which are significant risks to populations identified by most global health bodies.

## 2 COVID-19 epidemic growth rate stochastic regression models

In this section, we introduce a range of statistical population growth models that we will explore, in order to study the effect of model risk in COVID-19 epidemic modelling of the national-level cumulative number of infected individuals over time. In particular, the models we develop will be comprised of two components:

Observation Model: modelling the non-decreasing process for the cumulative count time-series for daily infections, and,

Latent State Model: modelling the stochastic trend in infection rate over time as a combination of three components: a graduation time trend component, regression factors that we introduce related to lagged effects of news sentiment from public health announcements, and a stochastic component to accommodate process uncertainty.

In this regard, we will formulate a class of regression models which is of the GLARMA type of time-series regression model for population growth, see examples of the development of such models in [15, 16]. However, in this work, we have two specifics present that differentiate our work from just standard application of classical population growth models in this context. The first is that we have adopted a flexible splice model for the observation model to accommodate two phases of the epidemic's evolution. The splice model allows us to capture the different waves of high infection rate intensity that has occurred over the year 2020 in which the epidemic has been occurring. A first significant infection wave in early January to April proceeded with an appeasement in infection rate through the summer of 2020, and then subsequently a second wave of infection, more substantial than the first, started to occur in many countries in Europe and the US towards the Winter of 2020 through November, and continues into March in 2021.

In order to capture these effects in a cumulative infection rate growth model, we have opted to use a class of splice observation models. In this manner, we can capture adequately both the infection dynamic at the start of the period of 2020, where zero-inflation was present as the infection was only just starting to achieve community transmission in various regions of each country, and later as the disease spread to a wider population transmission with the rapid rates of rise at the peaks of the transmission waves. These rates of rise result in a need for capturing over-dispersion features, and eventually, the cumulative counts were so substantial in most populations that a continuous approximation for the observation distribution is justified statistically as a model. Furthermore, the stochastic trend structure is developed for a variety of assumptions on growth rate behaviours. We consider the classical Gompertz model formulation as a reference to the more sophisticated models we developed using a combination of basis function regression, graduation temporal effects, and stochastic growth models for the trend in the cumulative daily new infections at the national level.

## 2.1 Splice observation model for national-level daily cumulative infections

In this section, we adopt a flexible observation distribution that aims to account for two or more driving processes that give rise to the observations of daily infection counts of COVID-19 at a national level, over time. The method known as splicing adopts different models for particular intervals of support, and in our case, we utilise this method to distinguish between early phase dynamics of the epidemic and the later widely mixing phase of the pandemic, the so-called community transmission regime. For a simple introduction to splice models, see [17].

In the cumulative count model setting, the splice model structure in turn naturally applies an implicit threshold-type effect on the models according to the observed new case counts daily magnitudes. To be precise, we model the cumulative daily new infections of COVID-19 counts at a national level over time. Therefore, small daily infection count observations may be modelled by one parametric model over a particular interval of time and cumulative observation magnitude. Then, when larger community transmission is taking place, our model will switch, naturally via the splicing structure, to a model more adequately able to capture large cumulative daily infection count observations. This second epidemic phase of infections is captured by the second mixture component of the splice model which is fitted directly to the

observations in the adjacent observation time partition, once community transmission has occurred significantly in a population. We also treat the splice time interval as part of the model selection exercise.

We note that we believe in this work for simplicity and ease of use by public health officials and decision-makers, therefore it is sufficient to consider two epidemic regimes. The first stage is the early stage spreading of the epidemic when the number of cumulative infections is low, which occurs at the start of the epidemic, where the number of infections satisfies that each day the count is below a selected threshold $y_*$ (to be estimated) as there is containment or isolated community transmission. Then the second stage of significant community transmission follows.

The corresponding density $f(y)$ and CDF $F(y)$ for the spliced observation model are

$$f(y) \quad = \omega f_1(y)\mathbb{I}_{y<y_*} + (1-\omega)f_2(y)\mathbb{I}_{y\geq y_*},$$

$$F(y) \quad = \begin{cases} \omega F_1(y), & 0 < y < y_* \\[2ex] \omega F_1(y_*) + (1-\omega)F_2(y), & y \geq y_*, \end{cases}$$

where $\omega \in [0, 1]$ is the weight parameter and the proper densities $f_1(y)$ and $f_2(y)$ (and their distribution functions $F_1(y)$ and $F_2(y)$) correspond to the densities $g_1(y)$ and $g_2(y)$ truncated above and below $y_*$, respectively:

$$f_1(y) = \frac{g_1(y)}{G_1(y_*)}\mathbb{I}_{y<y_*}, \;\; F_1(y) = \frac{G_1(y)}{G_1(y_*)}, \quad y < y_*,$$

$$f_2(y) = \frac{g_2(y)}{1 - G_2(y_*)}\mathbb{I}_{y\geq y_*}, \;\; F_2(y) = \frac{G_2(y) - G_2(y_*)}{1 - G_2(y_*)}, \quad y \geq y_*.$$

In the splice interval $[0, y_*)$ we will adopt a parametric distribution generically denoted by $F_1(y)$ with density $f_1(y)$ defined on $y < y_*$. In this study, we will consider for $f_1(y)$ a very flexible counting distribution given by a Generalised Poisson distribution. Once community transmission increases and the epidemic starts mixing steadily in the population on a wider basis, the cumulative number of infections over time will increase beyond the threshold and we will consider the support range $[y_*, \infty)$. In this case, since we are modelling the cumulative infections, this is equivalent to a regime-switching model type in the time threshold, but instead, we have captured the same idea via a splice model. Note this is due to the fact that we explicitly model cumulative infection counts daily over time. In this second phase of the epidemic, the adopted observation model uses a different parametric distribution, denoted generically by $F_2(y)$ with density $f_2(y)$ defined on $y \geq y_*$. This distribution will typically be a continuous distribution because the cumulative counts will be sufficiently large by this point that the discrete nature of the observation time-series can be adequately replaced by a continuous distribution approximation such as a normal observation model, or, in other words, normal distribution errors on the daily cumulative observations, conditional on a stochastic regression trend.

Next we outline specifically the models used for the observation model.

**Definition 2.1** (Observation Function). Given a discrete time-series process $\{N_t\}$ and a natural $\sigma$-algebra for the observed data filtration $\mathcal{F}_{1:t} = \sigma(N_1, E_1, N_2, E_2, \ldots, N_{t-1}, E_{t-1}, E_t)$, the

observation function for the time-series observation model is defined by

$$N_t | \mathcal{F}_{1:t} \sim F(n_t; E_t \phi(\eta_t), \quad v, \sigma) =$$

$$\begin{cases} \omega F_1(n_t; E_t \phi(\eta_t), v), & 0 < n_t < n_* \\ \\ \omega F_1(n_*; E_t \phi(\eta_t), v) + (1-\omega) F_2(n_t; E_t \phi(\eta_t), \sigma), & n_t \geq n_*. \end{cases}$$

where $N_t$ is the observed cumulative total infections count on day $t$, $E_t$ represents an exposure adjustment factor which may vary over time, and the latent trend for the intensity of the observation process that characterises the time-series regression trend is given by $\mu_t = E_t \phi(\eta_t)$, where $\phi(\cdot)$ is a suitably chosen link function, $\eta_t$ will denote the linear predictor for the trend, and $v$, $\sigma$ are parameters of $F_1$, $F_2$ respectively which are specified in Definition 2.2.

For the $g_1$ model we consider a flexible extension of the classical Poisson counting model given by the Generalised Poisson distribution, denoted by $GP(\eta, v)$, which has the probability mass function (p.m.f.), mean and variance given by Definition 2.2.

**Definition 2.2** (Generalised Poisson). A random variable $Y$ follows a Generalised Poisson distribution with support on $(\mathbb{N} \cup \{0\})$ if it has a p.m.f., mean and variance given respectively by

$$g_1(y; \eta, v) = \eta(1-v)[\eta(1-v) + vy]^{y-1} e^{-\eta(1-v)-vy}/y!, \;\; \eta > 0, \;\; -1 \leq v < 1,$$

$$\mathbb{E}(Y) = \eta \text{ and } \mathbb{V}\text{ar}(Y) = \eta(1-v)^{-2}.$$

The GP distribution is over-, under- and equi-dispersed when the dispersion parameter $v \in (-1, 1)$ is greater than, less than and equal to 0, respectively. For the second splice component, we will consider the $f_2$ density as a normal distribution given by $Y_t | Y_t \geq y_* \sim \mathcal{N}(\eta, \sigma)$. In both splice components $f_1$ and $f_2$ there is a parameter corresponding to the mean of the observation, denoted by $\eta$. In the next subsection, we will introduce the structure for $\eta$ which will allow us to specify the model structure for the growth in the trend over time, making it stochastic to obtain the functional form for $\eta_t$.

## 2.2 Latent stochastic growth model

The functional form of the trend in the observation of the regression splice model of Section 2.1 is specified by a variety of different stochastic trend models for the linear predictor $\eta_t$ (Eqs 1 and 2) that seek to capture the dynamic of the daily cumulative infections. We will universally adopt a link function $\phi(\cdot)$ given by the natural logarithm. Recall that our reference model for this linear predictor will be a Gompertz growth model [9] that we denote by model index M1 and the stochastic trend version of the Gompertz growth model is denoted by M2. We have introduced stochasticity in these models, which are usually found in a deterministic context, by adding a Gaussian noise process denoted by $\varepsilon_t$. Then, models M3-M6 are well-known population growth models: M3 is the Ricker model [18], M4 is the Theta-Logistic [19], M5 is the "mate-limited" logistic model [20] and M6 is the "Flexible-Allee" logistic model [21].

We make the following remarks about the behaviour of some of these models. With the Ricker equation (M3) model, the growth rate in cumulative infections will exhibit negative density dependence when $(b_1 < 0)$. The "carrying capacity" of the environment is defined by the stable equilibrium at $-\frac{\mu}{b_1}$ provided the density-independent growth rate is positive $(\mu > 0)$, otherwise $(\mu < 0)$, the only stable equilibrium is located at 0. While this model is linear in its parameters, it is non-linear in the latent state because it contains an exponential term in $N_{t-1}$. In the Theta-Logistic equation (M4), $b_2$ determines the form of density dependence. The

carrying capacity $\left(K = \left(-\frac{\mu}{b_1}\right)^{\frac{1}{b_2}}\right)$ exists provided $\mu$ and $b_1$ are of opposing sign, and it is stable only when $\mu$ and $b_2$ have the same sign. In the "mate-limited" logistic equation (M5), $b_3 > 0$ represents the population size at which the per-individual birth rate is half of what it would be if mating was unlimited, thus controlling the population size at which Allee effects are "noticeable". Only a strong Allee effect can be expressed by this model. For the "Flexible-Allee" logistic equation (M6), which allows for both strong and weak Allee effects, we obtain the roots $K = \frac{(-b_1 - \sqrt{b_1^2 - 4\mu b_4})}{2b_4}$ and $C = \frac{(-b_1 + \sqrt{b_1^2 - 4\mu b_4})}{2b_4}$. If these are real, then $C$ represents the threshold of population size below which the per capita population growth is negative. In a deterministic model without the noise process, when $0 < C < K$, the Allee effect is strong and an unstable equilibrium at $N = C$ may occur between two stable equilibria ($N = 0$ and $N = K$). When $C < 0$, the Allee effect is weak and a single stable equilibrium occurs at $N = K$. If $C$ and $K$ are not real, then $N = 0$ is the only stable equilibrium.

$$
\begin{aligned}
\text{M1} \quad & \eta_t = \ln(N_{t-1}) + \mu \exp(-b_0 t), \\[4pt]
\text{M2} \quad & \eta_t = \ln(N_{t-1}) + \mu \exp(-b_0 t) + \varepsilon_t, \\[4pt]
\text{M3} \quad & \eta_t = \ln(N_{t-1}) + \mu + b_1 N_{t-1} + \varepsilon_t, \\[4pt]
\text{M4} \quad & \eta_t = \ln(N_{t-1}) + \mu + b_1(N_{t-1})^{b_2} + \varepsilon_t, \\[4pt]
\text{M5} \quad & \eta_t = 2\ln(N_{t-1}) + \mu + b_1 N_{t-1} - \ln(b_3 + N_{t-1}) + \varepsilon_t, \\[4pt]
\text{M6} \quad & \eta_t = \ln(N_{t-1}) + \mu + b_1 N_{t-1} + b_4 N_{t-1}^2 + \varepsilon_t.
\end{aligned}
\tag{1}
$$

The models M7-M12 (Eq 2) are non-standard population growth models that we compare to models M1 to M6 as they provide additional degrees of freedom in order to capture inflection and turning points in cumulative infection rates over time. The additional structures include: in M7 the introduction of a stochastic log-linear growth combined additively with a Radial Basis Function growth component. The parameter $b_5$ in M7 determines the turning point of the convexity of the curve, which is also the fastest increasing point; M8 is a hybrid model between the stochastic Gompertz growth model and the Theta-Logistic growth model; M9 is a simple stochastic log-linear growth model consistent with a hypothesis of exponential infection rate growth over the long-term dynamics of the population; M10 and M11 are stochastic model variations of a basis function regression model widely considered in econometrics, known as the Nelson-Siegel model. This has not been previously considered in epidemic modelling, however, its flexible basis function structure offers a suitable structural representation for cumulative infection trend over time, and so we introduce this structure to the epidemic modelling literature in two forms: a state-dependent basis function form and a temporal trend dependence form. Finally, M12 is a model analogous to M7 with the square exponential radial basis function replaced by a Student-t density hyperbolic radial basis function that captures greater variation in the growth rates than model M7. By comparing these model structures to the calibration obtained with the Gompertz model we can assess model risk and then analyse how policy decisions based on infection rate forecasts could be affected

by this model risk.

$$\text{M7} \quad \eta_t = \ln(N_{t-1}) + \mu \exp\left(-\left(\frac{N_{t-1} - b_5}{b_6}\right)^2\right) + \varepsilon_t,$$

$$\text{M8} \quad \eta_t = \ln(N_{t-1}) + \mu \exp(-b_0 t) + b_1(N_{t-1})^{b_2} + \varepsilon_t,$$

$$\text{M9} \quad \eta_t = b_7 \ln(N_{t-1}) + \mu + \varepsilon_t,$$

$$\text{M10} \quad \eta_t = \ln(N_{t-1}) + \mu + b_8 \frac{1 - \exp\left(-\frac{N_{t-1}}{b_{10}}\right)}{\frac{N_{t-1}}{b_{10}}}$$

$$+ b_9 \left(\frac{1 - \exp\left(-\frac{N_{t-1}}{b_{10}}\right)}{\frac{N_{t-1}}{b_{10}}} - \exp\left(-\frac{N_{t-1}}{b_{10}}\right)\right) + \varepsilon_t, \qquad (2)$$

$$\text{M11} \quad \eta_t = \ln(N_{t-1}) + \mu + b_8 \frac{1 - \exp\left(-\frac{t}{b_{10}}\right)}{\frac{t}{b_{10}}}$$

$$+ b_9 \left(\frac{1 - \exp\left(-\frac{t}{b_{10}}\right)}{\frac{t}{b_{10}}} - \exp\left(-\frac{t}{b_{10}}\right)\right) + \varepsilon_t,$$

$$\text{M12} \quad \eta_t = \ln(N_{t-1}) + \mu\left(\frac{b_6^2}{(N_{t-1} - b_5)^2 + b_6^2}\right) + \varepsilon_t.$$

## 2.3 Incorporating the natural language signal in the model

The research work in [22] points out that classical epidemiological models do not consider that agents have an adaptive contact behaviour during epidemics. People, however, will change their behaviour based on society's sentiment regarding the epidemic (fear, panic, uncertainty etc.), and governmental policy measures taken to address the epidemic itself or resulting economic repercussions. Such behavioural changes will feed back into the spreading mechanism of the epidemic and potentially will have a significant impact on its evolution. It is therefore critical that one captures this epidemic- and behaviour-induced signal, and use it to inform epidemic models as a proxy to policy interventions and their impact.

In order to address this observation, we have added additional structure to each of the models previously presented. In particular, we will also add in a time-series distributed lag covariate based on a constructed time-series that we extract for the sentiment from news articles and public health announcements. The way in which we incorporate the sentiment index covariate is through an exposure adjustment to the link function transformed latent stochastic linear predictor process.

The motivation for incorporating such exposure adjustments based on public health announcements and news on the COVID-19 situation over time is to assess the role such

information has on the dynamics of new infection rates. The premise is that with clear and easy to follow public health guidance, the new infection rates should appease somewhat, especially in the early stages of the pandemic, as the reduction in uncertainty regarding how the disease spreads, what is safe practice and what is not, will help alleviate accidental transmissions or people being at higher risk, as they will supposedly adapt their behaviours according to the policy advice. This, in turn, should be captured and quantified by the sentiment index and then as an exposure adjustment, it will modulate the stochastic growth rate of cumulative infections.

In the Generalised Poisson model we are considering, the exposure variable sets a baseline level of disease counts which can be attributed to the imperfect testing, or misreporting. Adding the natural language component in the exposure to further inform that baseline level is therefore a reasonable choice, since news reports can express underlying expectations about the degree of disease spread in the community.

## 3 Bayesian model formulation and posterior predictive cumulative infection rates

In this section we show how to take the classes of epidemic growth rate models described in the previous section and incorporate them into a Bayesian model formulation. Then we demonstrate how to perform posterior inference on these models using an efficient Markov chain Monte Carlo (MCMC) package (STAN) that is widely used and runs in R and Python.

The Bayesian approach provides several advantages. Firstly, prior beliefs can be incorporated into model structures. Secondly, the Bayesian approach replaces the computation complexities in evaluating the marginal likelihood function, which involves high-dimensional integration of latent variables in the maximum likelihood (ML) approach, by posterior sampling. This advantage particularly applies to our proposed model because the model involves latent variables in the mean $\eta_t$. Thirdly, posterior predictive distributions provide distributional forecast summaries such as Bayesian prediction intervals. These intervals incorporate more sources of variability than the confidence intervals under the classical frequentist approach and are therefore often preferred.

In this section, we introduce the Bayesian approach to estimate the proposed models of Eqs 1 and 2. The basic idea of the Bayesian approach can be described by the following definition and equations.

**Definition 3.1** (Posterior distribution). Consider a set of observed values $N_t = (N_1, N_2, \ldots, N_T)$ with each $N_t \in (\mathbb{N} \cup \{0\})$, sentiment $E_{1:T}$ and the vector of unknown parameters

$$\boldsymbol{\vartheta} = (b_1, b_2, \cdots, b_{10}),$$

and denote the state parameters $\boldsymbol{\eta} = (\eta_1, \eta_2, \cdots, \eta_T)$ and

$$\boldsymbol{\vartheta}^* = [\boldsymbol{\vartheta}, \boldsymbol{\eta}].$$

The posterior distribution for $\boldsymbol{\vartheta}^*$ conditional on $N_{1:T}$ is given by

$$\pi(\boldsymbol{\vartheta}^*|\mathcal{F}_{1:T}) = \frac{f(N_{1:T}|E_{1:T}, \boldsymbol{\vartheta}^*) \ \pi(\boldsymbol{\vartheta}^*)}{\int f(N_{1:T}|E_{1:T}, \boldsymbol{\vartheta}^*) \ \pi(\boldsymbol{\vartheta}^*) \ d\boldsymbol{\vartheta}^*} \propto f(N_{1:T}|E_{1:T}, \boldsymbol{\vartheta}^*)\pi(\boldsymbol{\vartheta}^*),$$

where the prior densities $\pi(\boldsymbol{\vartheta})$ can be chosen based on available information or past data.

If one does not have a view or access to any a priori belief regarding the Bayesian model parameters, it is standard practice to utilise a class of non-informative or reference priors. Furthermore, the credible intervals for all parameters of interest can be constructed from posterior distributions. A credible interval is the Bayesian equivalent of the confidence interval in

frequentist statistics, but instead of being a random interval it is a deterministic quantile of the posterior or the posterior predictive distribution. In the case of the posterior, credible intervals capture our current uncertainty in the location of the parameter values and thus can be interpreted as a probabilistic statement about the parameter variability. In the case of the posterior predictive distribution, the credible intervals give us evidence of the posterior range of uncertainty regarding new predictions of cumulative infection rates that we have forecast from the model.

We can now formulate a generic form for the complete likelihood function for the splice model containing both the Generalised Poisson component and normal components, that will encapsulate the various model structures we employ, as follows:

$$
\begin{aligned}
f(\boldsymbol{N}_{1:T}|\boldsymbol{E}_{1:T},\boldsymbol{\vartheta}^*) \quad &= \prod_{t=1}^{T}(f_{GP}(N_t|E_t,\boldsymbol{\vartheta},\eta_t)\mathbb{I}_{y<y_*} + f_{\mathcal{N}}(N_t|E_t,\boldsymbol{\vartheta},\eta_t)\mathbb{I}_{y\geq y_*})f(\eta_t|\boldsymbol{\vartheta}) \\
&= \prod_{t=1}^{T}\frac{E_t\phi(\eta_t)(1-v)}{\Gamma(N_t+1)\exp(N_t v)}\frac{[E_t\phi(\eta_t)(1-v)+N_t v]^{N_t-1}}{\exp\{E_t\phi(\eta_t)(1-v)\}} \\
&\quad \times \frac{1}{\sqrt{2\pi\sigma^2}}\exp\left(-\frac{N_t-E_t\phi(\eta_t)}{2\sigma^2}\right) \times \frac{1}{\sqrt{2\pi\sigma_\varepsilon^2}}\exp\left(-\frac{\varepsilon_t^2}{2\sigma_\varepsilon^2}\right),
\end{aligned}
\tag{3}
$$

where $f(\eta_t)$ are the different structures to model various responses (Eqs 1 and 2). The following priors $\pi(\boldsymbol{\vartheta}_x)$ are adopted in this paper

$$
v \sim \mathrm{U}(-1,1), \quad \sigma^2 \sim \mathrm{Gamma}(1,1), \quad \sigma_\varepsilon^2 \sim \mathrm{Gamma}(1,1), \quad b_i \sim \mathcal{N}(0,1), \quad i \in (1,\dots,10),
$$

where $\mathrm{U}(a_\theta, b_\theta)$ denotes the uniform priors on the range $(a_\theta, b_\theta)$ for parameter $\theta$ which represents the shape parameter $v$. In time-series and regression settings, it is fairly common to use a normal distribution for coefficients. The model coefficients $b_1, b_2, \cdots, b_{10}$ follow a normal distribution with mean equal to 0 and variance equal to 1. The choice of the mean was informed by empirical studies. Moreover, setting the variance to 1 allows for great flexibility with regard to this mean specification and makes the prior relatively uninformative when viewed on the log-scale. $\mathrm{Gamma}(a, b)$, denoting the gamma prior with shape and scale parameters $a$ and $b$, is an adequate choice for positive variances $\sigma_\varepsilon^2$.

One outstanding advantage of Bayesian inference in forecasting is the construction of posterior predictive distributions for all forecasts. In this study, $m$-step forecasts $\boldsymbol{N}_{T+1:T+m}$ are constructed via a sequence of 1-step ahead forecasts of $N_{T+s}$, $s = 1, \dots, m$ using a sliding window, where the observed data filtration for each window is $\mathcal{F}_{s:T+s-1}$. The posterior predictive distribution for $N_{T+s}$, $s = 1, \dots, m$ is defined as

$$
\begin{aligned}
&f(N_{T+s}|\mathcal{F}_{s:T+s-1}) \\
&= \int \cdots \int f(N_{T+s}|\eta_{T+s},\boldsymbol{\vartheta},\mathcal{F}_{s:T+s-1})f(\eta_{T+s}|\boldsymbol{\eta}_{s:T+s-1},\boldsymbol{\vartheta},\mathcal{F}_{s:T+s-1})f(\boldsymbol{\vartheta}|\mathcal{F}_{s:T+s-1})\ d\boldsymbol{\eta}_{s:T+s}d\boldsymbol{\vartheta},
\end{aligned}
$$

and this integral can be approximated by the Monte Carlo estimator, constructed from posterior samples according to:

$$
\hat{f}(N_{T+s}|\mathcal{F}_{s:T+s-1}) = \frac{1}{L}\sum_{l=1}^{L} f(N_{T+s}|\boldsymbol{\eta}_{s:T+s}^{(l)},\boldsymbol{\vartheta}_s^{(l)},\mathcal{F}_{s:T+s-1}).
$$

In this study, we set $L = 90{,}000$ as the number of iterations after burn-in in each MCMC sampler run given the current window of information $\mathcal{F}_{s:T+s-1}$. In addition, $\boldsymbol{\eta}_{s:T+s}^{(l)}$ and $\boldsymbol{\vartheta}_s^{(l)}$ are the $l$-

th draw in the posterior sample of $\boldsymbol{\eta}_{s:T+s}$ and $\vartheta$, respectively. For Bayesian inference, apart from the posterior predictive distribution, the various posterior predictive point estimators and predictive credible intervals can also be obtained. Empirical Bayes forecasts as a typical forecast estimator in the Bayesian setting can improve the computational efficiency [23]. For empirical Bayes forecasts, the calculations undertake conditional upon selected posterior (in-sample) point estimators denoted by $\tilde{\boldsymbol{\vartheta}}$ and $\tilde{\boldsymbol{\eta}}_{s:T+s}$, rather than integrating out posterior (in-sample) parameter uncertainty from the predictive distribution and resultant forecast estimators [24]:

$$\hat{f}^{EB}(N_{T+s}|\mathcal{F}_{s:T+s-1}) = f(N_{T+s}|\tilde{\boldsymbol{\eta}}_{s:T+s}, \tilde{\boldsymbol{\vartheta}}, \mathcal{F}_{s:T+s-1}).$$

Typically, the point estimators used in $\tilde{\boldsymbol{\vartheta}}_s$ (similarly for $\tilde{\boldsymbol{\eta}}_{s:T+s}$) are either formed from the maximum-a-posteriori estimate (MAP) or the estimate which minimises the Posterior Expected Loss (PEL). The concept of MAP is similar to the ML estimate when the priors are uninformative since in this case $\tilde{\boldsymbol{\vartheta}}_s$ is the mode of the posterior distribution

$$\tilde{\boldsymbol{\vartheta}}_{s,MAP} = \arg\max_{\boldsymbol{\vartheta}_s} f_{\boldsymbol{\vartheta}_s}(\boldsymbol{\vartheta}_s|\mathcal{F}_{s:T+s-1}).$$

Alternatively, the Bayes estimator which minimises the Posterior Expected Loss is defined as

$$\tilde{\boldsymbol{\vartheta}}_{s,PEL} = \arg\min_{\boldsymbol{\vartheta}_s} \mathbb{E}[L(\boldsymbol{\vartheta}_s, \tilde{\boldsymbol{\vartheta}}_s|\mathcal{F}_{s:T+s-1})],$$

where $L(\boldsymbol{\vartheta}_s, \tilde{\boldsymbol{\vartheta}}_s|\mathcal{F}_{s:T+s-1})$ is the loss function. One example is the commonly used minimum mean square error (MSE) estimator defined as

$$\tilde{\boldsymbol{\vartheta}}_{s,MSE} = \arg\min_{\boldsymbol{\vartheta}_s} \mathbb{E}[(\boldsymbol{\vartheta}_s - \tilde{\boldsymbol{\vartheta}}_s)^2|\mathcal{F}_{s:T+s-1}],$$

where $\tilde{\boldsymbol{\vartheta}}_{s,MSE}$ corresponds to the posterior mean $\mathbb{E}[\boldsymbol{\vartheta}_s|\mathcal{F}_{s:T+s-1}] = \bar{\boldsymbol{\vartheta}}_s$. If the minimum absolute error (AE) estimator $L(\boldsymbol{\vartheta}_s, \tilde{\boldsymbol{\vartheta}}_s) = |\boldsymbol{\vartheta}_s - \tilde{\boldsymbol{\vartheta}}_s|$ is used, it gives $\tilde{\boldsymbol{\vartheta}}_{s,AE} = \boldsymbol{\vartheta}_{s,0.5}$ which is the posterior median.

## 4 Data and sentiment index construction methodology

In this section we describe the variety of data sources collected for both the cumulative daily level of infection rates of COVID-19 by national statistics in a range of countries, as well as the public health information announcements and news articles on public health warnings, policy and guidance published. We restricted to English language news and information sources to avoid any ambiguity that may arise in translation effects if other language news sites were introduced and had to be translated for sentiment extraction.

### 4.1 COVID-19 national-level daily infection counts

At the national-level statistics, we have to select between three basic sources of data that could be studied: daily new COVID-19 infections, daily death counts from COVID-19 or daily recovery counts. We had to consider which source of data would be most reliable to study and most applicable to the class of models considered and the introduction of the sentiment exposure indicator. After preliminary analysis of numerous widely available data sources for such national-level statistics in a variety of countries, we determined that we would work with the number of daily infections that we turned into a cumulative count of infections over time. We model the cumulative number of infected cases as we consider it to be the most reliable

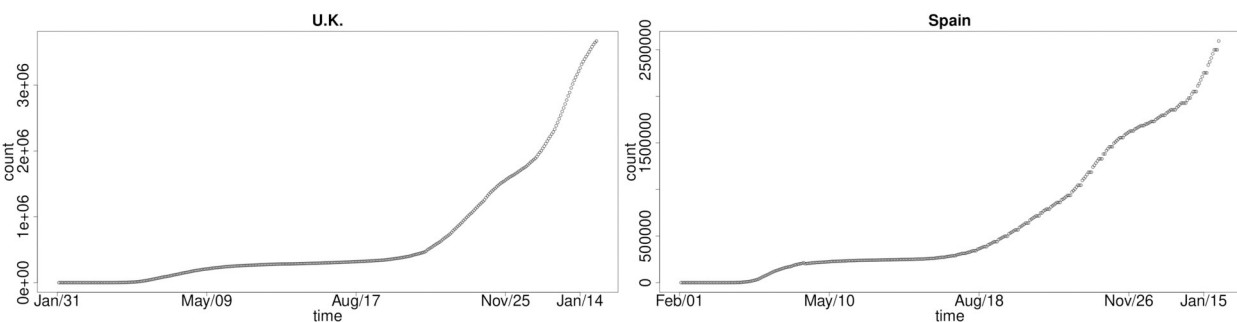

**Fig 1. Cumulative COVID-19 national-level infections over time: UK (left) and Spain (right).**

summary statistic of the spread of COVID-19. The number of patients who recovered is not updated with the same frequency and rigour as the number of infected or deceased cases. Regarding the death counts, there are well-known challenges with using this data as numerous grouping, clustering and misreporting adjustments related to the death counts for COVID-19 have been widely reported. Furthermore, the number of deaths and how they are reported in relation to attribution to COVID-19 death or other complications and the schedule of releasing such data varied widely with countries.

The converse issue with modelling the number of infections daily is that it is also the case that the testing efficacy was not always very high with many tests proving faulty, especially early in the epidemic. Despite this, we will proceed with the daily infection data and assume that the stochasticity introduced in the model infection rate will reasonably account for this uncertainty in the true observations arising from diversity in testing practice, testing types etc.

We focus on the following seven countries: Australia (AU), Germany (GM), Italy (IT), Japan (JP), Spain (SP), United Kingdom (Wales, England and Scotland—UK) and United States (US). We collected data for the period January 2020 to the middle of February 2021. The source of our data is the COVID-19 Data Repository by the Centre for Systems Science and Engineering (CSSE) at Johns Hopkins University (https://coronavirus.jhu.edu/).

We plot the number of COVID-19 infected cases for the seven selected developed countries in order to get a perspective on the growth curve structures we will seek to explore with our models. For all seven countries, there exists a steep increment after the end of August 2020. This wave of the outbreak could be caused by the seasonal change. We have grouped them according to the basic structures they present for the shape of the cumulative COVID-19 infections. We see that in Figs 1 and 2 we have common national-level infection growth dynamics which are consistent in structure for the UK, Spain, Italy and Germany. In these countries, the

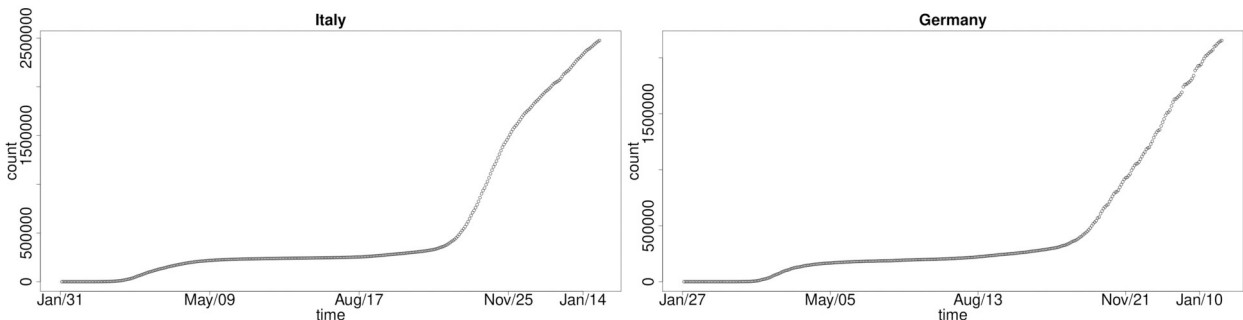

**Fig 2. Cumulative COVID-19 national-level infections over time: Italy (left) and Germany (right).**

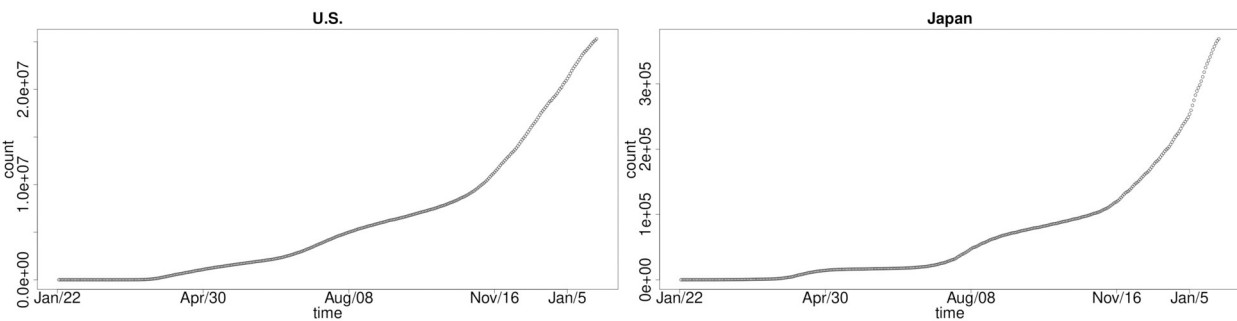

**Fig 3. Cumulative COVID-19 national-level infections over time: US (left) and Japan (right).**

number of infected cases in the first wave increased in similar fashion in the early stages of the pandemic with a rapid rise in new cases to a significant infected population, followed by a slowdown in national-level cumulative infections as government policies and public aware-ness, testing campaigns and lockdown measures took effect. Through mid May 2020 to around mid August 2020 there was a clear stabilisation and significant reduction in growth rate of new infections. Then once the second and third waves occurred we have seen sudden steep growth in the national-level cumulative infections in these countries of a similar structure and growth rate.

This is distinct from the epidemic's evolution in the US and Japan, demonstrated in Fig 3. In the US, we have not seen the punctuated clearly delineated phases of wave 1, wave 2 and wave 3 of the cumulative number of cases, rather we have seen a sequence of increasing growth rates in cumulative infections which have the same relative growth structure but at different total magnitudes. That is, these countries have experienced less of a pronounced decline in infection growth rate between each wave of infection of COVID-19.

Finally, in Fig 4 we see the plot for cumulative infections for Australia, where like the UK and the EU countries there exist three clear stable levels which correspond to three waves of infection growth. However, there are distinctive features in the Australian experience related to the fact that the interarrival time between each wave of infection is longer than in the UK

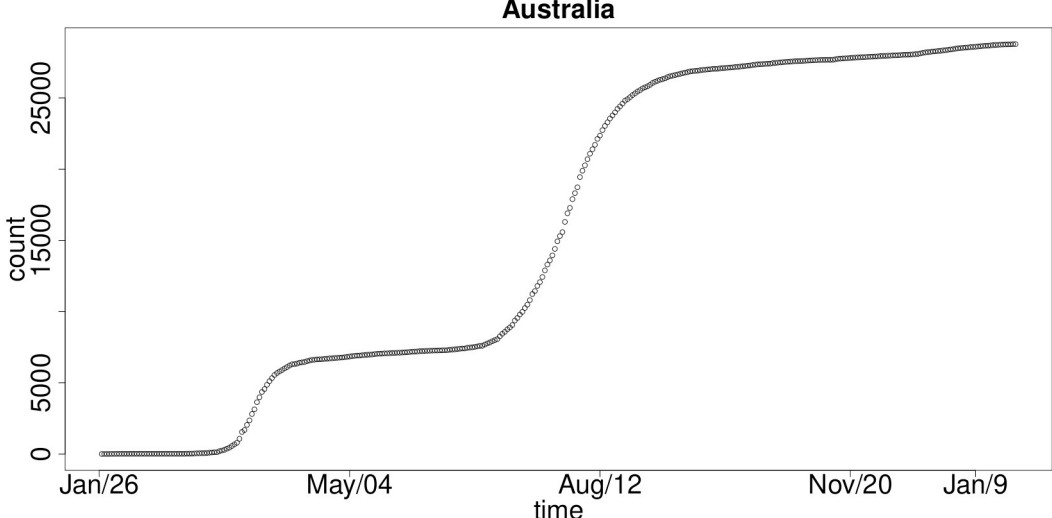

**Fig 4. Cumulative COVID-19 national-level infections over time: Australia.**

and the EU, and furthermore, the growth rates of infection are commensurate between each wave indicating a different pattern. Whilst each subsequent wave of infection in the UK and the EU were increasingly worse in both growth rate and total infection counts, the Australian experience was relatively consistent in magnitude and growth rates in each wave of infection.

These aspects of the growth rate dynamics for national-level cumulative infections are important to consider as they will have a significant effect on the adequacy of the model selected to capture such dynamics and could manifest in model risk and inaccurate decision-making if a one-size-fits-all model such as a Gompertz model were applied to try to capture such dynamics as we will demonstrate.

Because of their similar growth dynamics with the rest of the studied countries, in this manuscript we will present results for the UK, Germany, the US and Australia, and include the rest of our analyses in Sections B-E in S1 Appendix.

## 4.2 COVID-19 natural language processing text data

In addition to modelling the count data of infected cases, we also collected and processed a dataset of text documents composed of public news articles and health announcements related to COVID-19. These were collected from both high-circulation newspapers with careful editorial process, as well as press releases of public disease control institutions in Europe and the United States. The period of collected data is the months from November 1, 2019 to early August 2020, when the pandemic was at its start and we expect that news reporting will clearly reflect the strong sentiment present in the society. A summary of the data sources and related details is presented in Table 1 and Fig 5. Some of the sources provided a selection of articles that were already restricted to COVID-19, while others, mainly the Centres for Disease

**Table 1. Details on collected articles (after post-processing).**

| Source | Total #articles | Total #tokens | Total #sentences | Median of #tokens in article | Median of #sentences in article | Time period | Keywords |
|---|---|---|---|---|---|---|---|
| **European Centre for Disease Prevention and Control** | 12 | 2,510 | 212 | 184 | 16 | 01/11/2019 —06/08/ 2020 | coronavirus, COVID19, nCoV, COVID-19, epidemic, pandemic, COVID-19, epidemic, pandemic, Italy, Spain, Greece |
| **The New York Times United Nations Economic** | 10,544 | 4,965,016 | 547,989 | 414 | 45 | 21/01/2020 —06/08/ 2020 | categorised by source |
| **Commission for Europe** | 52 | 17,508 | 1,287 | 304 | 23 | 23/03/2020 —31/07/ 2020 | categorised by source |
| **United States Centres for Disease Control and Prevention** | 55 | 34,345 | 5,387 | 238 | 22 | 27/01/2020 —30/07/ 2020 | coronavirus, COVID-19 |
| **World Health Organisation** | 555 | 182,821 | 16,895 | 291 | 25 | 09/01/2020 —06/08/ 2020 | categorised by source |
| **The Telegraph** | 11,918 | 3,289,002 | 343,226 | 247 | 24 | 23/01/2020 —06/08/ 2020 | categorised by source |
| **The Guardian** | 13,719 | 4,960,699 | 515,643 | 335 | 32 | 23/01/2020 —06/08/ 2020 | categorised by source |
| **Reuters (blog)** | 211 | 64,306 | 6,204 | 239 | 23 | 12/03/2020 —06/08/ 2020 | categorised by source |

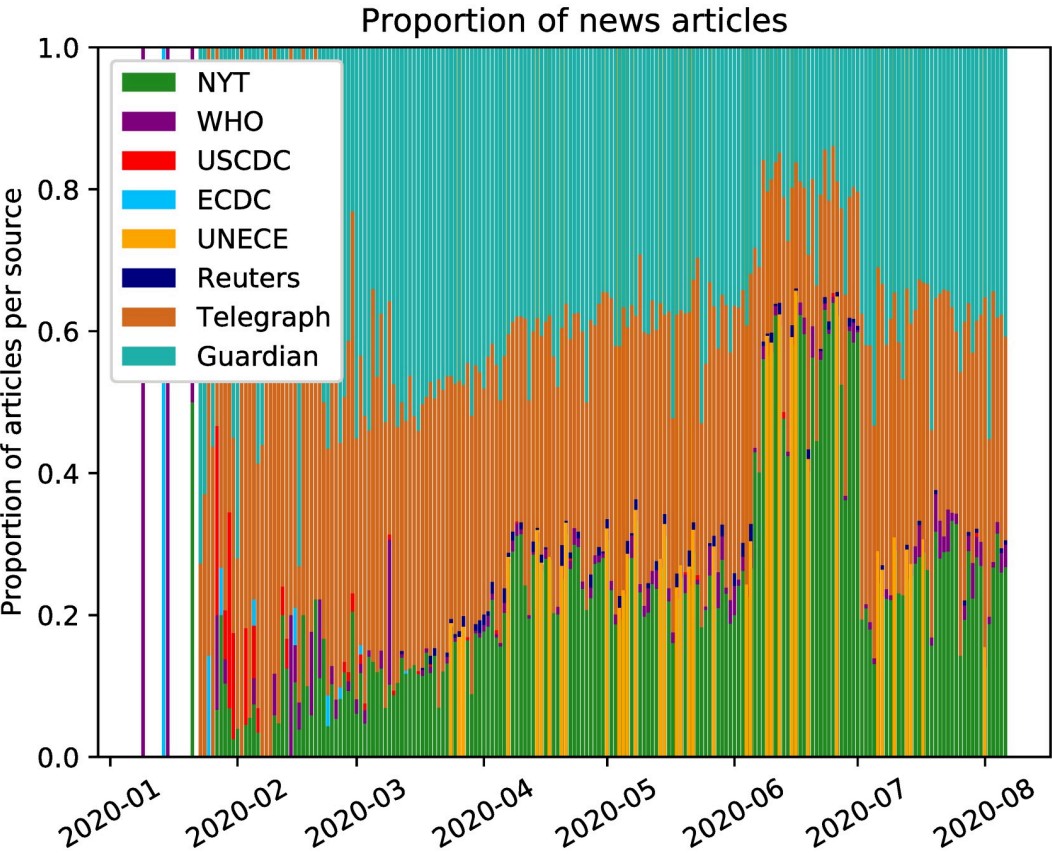

**Fig 5. Proportion of news articles.** Proportion of volume of news reports per news source.

Control, provided reporting on multiple diseases for the same period. When that was the case, we filtered the articles according to a selection of keywords that we include in Table 1.

Applying natural language processing statistical methods we aim to capture from the included news reports information about the way the pandemic and the governmental and national public health centres countermeasures have affected people's life: what is the effect on the economy, unemployment, travel industry, cultural and sports events, as well as personal well-being and psychological health. All these factors will inherently reflect people's reaction to the pandemic and, to some extent, will influence the manner and degree to which people may adhere to governmental protective advice. It is therefore meaningful to include this information in our predictive models. We note, that such news are affected by the quality of reporting and the characteristics (geographical, political, pertinent to educational level) of the target audience that the news sources address. We are therefore careful to choose news sources that are widely accepted to deliver high-quality reporting.

In terms of geographical characteristics, our richest news sources apart from health institutions (The New York Times, The Telegraph, The Guardian, Reuters blog) are US-, UK-, and Europe-based, however, they do attract a worldwide audience, at least in countries where English is amongst the official languages. Therefore, we will assume that the sentiment inherent in the articles is representative for part of the public sentiment in all of the countries of our study.

In constructing the dataset we wrote custom `Python` scripts to extract the text of the articles from the online site of each source. We did not store images, tables, figures or lists that

might have been included in some of the articles. We performed a text cleaning and pre-processing stage, where we remove the noise, in terms of unwanted characters, that is present after the collection of the articles. This is important to allow us to capture the useful statistical structure of text.

The following subsections outline key components of our sentiment signal extraction, detailing how we defined the sentiment index, and what our source of reference was for determining sentiment strength. We note that we did not attempt to classify sentiment polarity (positive, negative or neutral), as some sentiment modelling approaches do, since in this particular case it is highly dependent on perspective. Instead, to avoid this issue in the challenging COVID-19 context, we chose to quantify sentiment strength via an entropy measure, with low values of the sentiment index reflecting weak public attention on the public health announcements presented, and high sentiment values reflecting strong attention and therefore more likely adherence to such policy announcements or health guidance from the news audience.

**4.2.1 Reference dictionary.** In this work, we adopt a lexicon-based approach to extract sentiment from text. In lexicon-based sentiment modelling settings one must construct or work with a reference word dictionary that acts as a basis upon which all data is related, in this case words, termed tokens going forward. In this section, we will describe the need for us to construct such a dictionary for this specific application.

Contrary to the common approach of constructing sentiment lexicons, i.e., collections of words related to a specific sentiment (positive, negative, neutral), and then quantifying sentiment as some function of the number of words of a specific sentiment present in the text, we cannot apply this approach for news regarding the spread and impact of COVID-19. This is because it is very hard to classify words as expressing a certain sentiment in the context of our application, where news and the selection of words can be classified as positive or negative depending on one's perception of the pandemic evolution, political beliefs or personal views on the way the pandemic is being handled. For example, most people realised that the imposed quarantines were critical to contain the spread of COVID-19, but at the same time, restricting the freedom of movement is something that everyone would rather avoid. Therefore, there have been many reports in news about the positive impact of quarantines on protecting the people and the health systems, but at the same time many articles have been cautioning against their repercussions on the economy and personal health.

To address this challenge, first, we collect dictionaries that are specifically related to epidemic modelling, politics, business and psychology, as all of these are topics related to the articles that cover COVID-19 and its impact. It is important to remark here that often people construct dictionaries via collecting the most frequent tokens present in the corpus of documents that is available for training and evaluation of their model, yet we argue that this approach significantly restricts the representational power of the dictionary. In contrast, we separated the construction of the dictionary from the available text data. The dictionaries were constructed by collecting words present in online dictionaries, mainly those of Oxford University, in addition to online word lists that we identified as relevant to the topics of interest. All sites we used to obtain the dictionaries are documented in Table 2. After obtaining the word lists via web scraping, we further curated them by cleaning the tokens from scraping artefacts. When processing the news articles, any terms that were not part of the dictionaries of Table 2 were removed and their percentage per news source is documented in Table 3. Secondly, we construct a text time-series of the distribution of proportions of dictionary tokens in segments of text in the online time-dependent fashion we present in Section 4.2.2. Finally, we construct a sentiment index that quantifies sentiment as the dispersion of this distributional time-series of proportions.

**Table 2. Details of the sources we used to build our dictionary.** Dictionary size is measured in number of words.

| Dictionary | Size | Source |
|---|---|---|
| Epidemics | 48,657 | https://www.oxfordreference.com/view/10.1093/acref/9780199557141.001.0001/acref-9780199557141 |
| | | https://www.oxfordreference.com/view/10.1093/acref/9780195314496.001.0001/acref-9780195314496 |
| | | https://www.oxfordreference.com/view/10.1093/acref/9780198529170.001.0001/acref-9780198529170 |
| | | https://www.oxfordreference.com/view/10.1093/acref/9780199204625.001.0001/acref-9780199204625 |
| | | https://www.oxfordreference.com/view/10.1093/acref/9780198568506.001.0001/acref-9780198568506 |
| | | https://www.oxfordreference.com/view/10.1093/acref/9780199211777.001.0001/acref-9780199211777 |
| | | https://www.oxfordreference.com/view/10.1093/acref/9780198609957.001.0001/acref-9780198609957 |
| Philosophy and Psychology | 11,875 | https://www.oxfordreference.com/view/10.1093/acref/9780198794790.001.0001/acref-9780198794790 |
| | | https://www.oxfordreference.com/view/10.1093/acref/9780198735304.001.0001/acref-9780198735304 |
| | | https://www.oxfordreference.com/view/10.1093/acref/9780199657681.001.0001/acref-9780199657681 |
| | | https://www.collinsdictionary.com/word-lists/philosophy-philosophical-schools-and-doctrines |
| | | https://www.collinsdictionary.com/word-lists/psychology-psychology-terms |
| Politics and Government | 2,450 | https://www.oxfordreference.com/view/10.1093/acref/9780199670840.001.0001/acref-9780199670840 |
| | | http://www.english-for-students.com/Politics-Vocabulary.html |
| | | https://www.excellentesl4u.com/esl-politics-vocabulary.html |
| | | https://www.scholastic.com/teachers/articles/teaching-content/vocabulary-political-words/ |
| | | https://www.macmillandictionary.com/thesaurus-category/british/general-words-relating-to-politics-and-government |
| | | https://www.vocabulary.com/lists/183710 |
| | | https://myvocabulary.com/word-list/politics-vocabulary/ |
| | | https://www.collinsdictionary.com/word-lists/government-types-of-government |
| Business, Economics, and Finance | 86,545 | When is a Liability not a Liability? Textual Analysis, Dictionaries, and 10-Ks [25] |
| | | https://www.collinsdictionary.com/word-lists/economics-branches-of-economics |
| Baseline dictionary | 245,318 | Academic wordlist: https://www.wgtn.ac.nz/lals/resources/academicwordlist |
| | | Oxford 5000 word list:https://www.oxfordlearnersdictionaries.com/wordlists/oxford3000-5000 |
| | | Oxford World-Place names:https://www.oxfordreference.com/view/10.1093/acref/9780199580897.001.0001/acref-9780199580897 |
| | | https://www.rollingstone.com/music/music-lists/100-greatest-artists-147446/ |
| | | https://www.biographyonline.net/people/100-most-influential.html |
| | | https://time.com/collection/most-influential-people-2018/ |
| | | https://time.com/collection/100-most-influential-people-2019 |
| | | Unix built-in default dictionary |

The choice of dictionary (Table 2) when constructing the text time-series and the sentiment index will determine both the richness of representation of the embedding and the expressive power of the sentiment index. A poor dictionary without variability will lead to modelling runs (sequences) of zeros when constructing the time-series from an input text. In addition,

**Table 3. Percentage of out-of-vocabulary words.**

| Source | OOV |
|---|---|
| ECDC | 2.7% |
| NYT | 4.3% |
| UNECE | 3.6% |
| USCDC | 4.2% |
| WHO | 4.5% |
| The Telegraph | 3.99% |
| The Guardian | 12.1% |
| Reuters (blog) | 6.7% |

when interpreting the results based on the sentiment index, it is more likely to avoid being mislead by capturing non-relevant semantics if the dictionary contains many commonly used words within the application context. Therefore, the reliability of the dictionary sources, and the task domain become crucial when constructing the dictionary. To ensure reliable and up-to-date dictionaries, we used Oxford's dictionaries of English on the topics specified: epidemic modelling, politics, business and psychology, as detailed in Table 2. The application domain becomes especially important considering that words appear with different meanings and varying frequency in different contexts, and this has to be accounted for when interpreting the structural properties of the time-series.

**4.2.2 Construction of time-series of distributions via sequential text data embedding.** In this section we introduce how to transform the processed text tokens into a time-series of distributions, in the process explaining what is known in the NLP context as the text embedding representation.

Note that we are specifically interested in producing text embeddings with the aim to incorporate them in time-series regression models. In terms of literature, not many approaches have been developed for that purpose and in a sophisticated enough manner such that the produced time-series are useful for the type of processing we want to achieve. Such approaches include simple constructions based on letter and (global, non time-dependent) token counts or space-filling curves [26–28]. Furthermore, also few approaches are constructing sentiment indices in a way that allows for time-series type modelling. A recent example is [29], who construct a sentiment scoring rule based on the difference between the number of positive and negative words in Tweets, which is an approach significantly different to ours as we explained in the previous section.

The embedding framework we construct is based on the widely used *bag-of-words model* (BoW), which is commonly applied in natural language processing (NLP) and information retrieval [30]. The idea behind BoW in NLP is to represent a segment of text as a collection ('bag') of unordered words. We are now setting BoW into a time-series context, and present a novel online formulation that allows us not only to overcome computational difficulties associated with BoW, but also to incorporate the text-based sentiment index into our time-series system.

We begin by introducing some basic notation: $t$ denotes a 'token', i.e. a linguistic unit of one or more characters (a word, a number, a punctuation character etc), $\mathcal{V}$ is the *vocabulary*, i.e. a finite set of tokens that is acceptable by the language, and $\mathcal{D}$ is a *dictionary* ($\mathcal{D} \subseteq \mathcal{V}$), i.e. a finite set of tokens, which we consider expressive and relevant to the topic under study. We will work with $n$-grams, where $n$ denotes the number of tokens in the text processing unit we consider, namely a set of $n$ consecutive terms.

The time-series embedding is defined by the 3-ary relation $\mathcal{R} \subseteq \mathcal{V} \times \mathbb{D} \times \tilde{\mathcal{N}}$, where $\mathbb{D} = \{\mathcal{D}^1, \mathcal{D}^2, \ldots, \mathcal{D}^p\}$, $\mathcal{D}^j \subseteq \mathcal{V}$ is a set of dictionaries each of size $q_j$, and $\tilde{\mathcal{N}} = \{\mathbb{N}^{q_1}, \mathbb{N}^{q_2}, \ldots, \mathbb{N}^{q_p}\}$. To compute the members of $\tilde{\mathcal{N}}$ for each element of $\mathcal{R}$ we use the following equation, which defines $\mathcal{R}$:

$$\hat{\gamma}_N^{j,l}(\tilde{v}_N, \mathcal{D}^{j,l}) = \begin{cases} \dfrac{m_N^{j,l}}{n \times N}, & r_m(\tilde{v}_N, \mathcal{D}^{j,l}) = 1 \\ 0, & \text{otherwise,} \end{cases} \tag{4}$$

where $\tilde{v}_N = \{\tilde{v}_{wN}\}_{w=1:n}$, $m_N^{j,l} = |\{v' : v' \in \tilde{v}_N\} \cap \{\mathcal{D}^{j,l}\}|$, $\mathcal{D}^{j,l}$ denotes a dictionary token $l \in \{1,$

. . ., $q_j$}, for dictionary $j \in \{1, \ldots, p\}$, and

$$r_m\left(\tilde{v}_N, \mathcal{D}^{j,l}\right) = \begin{cases} 1, & m_N^{j,l} \geq m_{min} \\ \\ 0, & \text{otherwise,} \end{cases} \tag{5}$$

where $N$ is the index of the current timestep, in $n$-gram 'time' (time here indexes $n$-grams). Therefore, at each $N$ we have a vector of dimension $q_j$ which is the embedding of the $n$-gram at $N$. In this construction, the condition in Eq 5 restricts the count of any token of $\mathcal{D}^j$ which is in $n$-gram $v_{1N}, \ldots, v_{nN}$ at timestep $N$ to be at least $m_{min}$.

In order to capture the time-dependent nature of text, we note that the total number of observed tokens increases as we shift the $n$-gram towards the end of the text. Therefore, we want to recursively extract proportions of the dictionary tokens within the $n$-gram at time $N$. To account for this effect we apply the following transformation at each $N$:

$$\tilde{\hat{\gamma}}_N^{j,l}(\cdot) = \begin{cases} \dfrac{\sum_{i=1}^{N-1} m_i^{j,l} + m_N^{j,l}}{M_N}, & r_m(\cdot) = 1 \\ \\ 0, & \text{otherwise,} \end{cases} \tag{6}$$

where $m_N^{j,l}$ is the count of token $l$ in dictionary $\mathcal{D}^j$ at timestep $N$, and $M_N$ is the total count of tokens we have observed up to timestep $N$ which satisfy $r_m(\cdot) = 1$.

It is important to point out at this stage that the support of the distribution of proportions is restricted by the condition in Eq 5. Tokens with count less than $m_{min}$ will be excluded from $M_N$, and consequently the support of the distribution. To construct the time-series for the current study, we set $n = 20$ and $m_{min} = 1$.

**4.2.3 From distributional time-series to sentiment index.** The final stage of the construction then involves mapping this time-series of distributions onto a scalar summary to create a sequence of summary statistics that will define the sentiment index time-series.

Using the embedding extracted from token occurrences, we construct additional time-series using properties of the empirical distribution of the embedded text. We acquire the density of the token proportions of Eq 6:

$$g_N^{j,l}\left(\tilde{v}_N, \mathcal{D}^{j,l}\right) = \frac{\mathbb{I}^{j,l}(\tilde{v}_N)\tilde{\hat{\gamma}}_N^{j,l}\left(\tilde{v}_N, \mathcal{D}^{j,l}\right)}{\sum_{l=1}^{q_j} \mathbb{I}^{j,l}(\tilde{v}_N)\tilde{\hat{\gamma}}_N^{j,l}\left(\tilde{v}_N, \mathcal{D}^{j,l}\right)}, \tag{7}$$

where, as before, $\tilde{v}_N$ denotes the $n$-gram at time-step $N$, and the indicator function $\mathbb{I}^{j,l}(\tilde{v}_N)$ selects the $n$-gram terms:

$$\mathbb{I}^{j,l}(\tilde{v}_N) = \mathbb{I}(\tilde{v}_N, \mathcal{D}^{j,l}) = \begin{cases} 1, & \text{if } l \in \{l' \ : \ \mathcal{D}^{j,l'} \in \tilde{v}_N\} \\ \\ 0, & \text{otherwise,} \end{cases} \tag{8}$$

and then we can effectively study the density itself, that changes per $n$-gram, or use a suitable summary of it.

We expect that the frequency with which words are used in the course of the text, as well as the richness of the dictionary, will reflect on the value of the entropy of the empirical distribution of proportions, which we use to construct our time-series. The entropy is a vector-valued process of dimension $p$, $\boldsymbol{H}_N = [H_N^{(1)}, \ldots, H_N^{(p)}]$, whose marginal component that corresponds

**Table 4. Median monthly sentiment per news source and estimated 95% confidence intervals.** Dashes denote lack of published articles during those months.

| News source | January | February | March | April | May | June | July | August |
|---|---|---|---|---|---|---|---|---|
| ECDC | 2.313 ± 6.46E-02 | 2.646 ± 6.22E-02 | 2.267 ± 4.53E-02 | - | 2.422 ± 2.86E-02 | 2.296 ± 5.23E-02 | - | - |
| NYT | 2.265 ± 2.16E-02 | 2.043 ± 3.99E-03 | 1.987 ± 1.46E-03 | 1.979 ± 1.08E-03 | 1.976 ± 1.23E-03 | 1.996 ± 1.10E-02 | 2.012 ± 1.53E-03 | 2.016 ± 3.35E-04 |
| UNECE | - | - | 2.648 ± 1.62E-02 | 2.425 ± 4.63E-03 | 2.257 ± 1.64E-02 | 2.322 ± 9.81E-03 | 2.330 ± 9.78E-03 | - |
| USCDC | 2.380 ± 1.11E-02 | 2.221 ± 4.17E-03 | 2.180 ± 1.76E-02 | 2.088 ± 2.88E-02 | 2.060 ± 1.50E-03 | 2.109 ± 5.75E-03 | 2.166 ± 1.20E-02 | - |
| WHO | 2.684 ± 2.61E-01 | 2.401 ± 1.42E-02 | 2.174 ± 6.64E-02 | 2.123 ± 5.10E-03 | 2.098 ± 4.17E-03 | 2.072 ± 5.65E-03 | 2.090 ± 5.05E-03 | 2.104 ± 9.13E-03 |
| Telegraph | 2.186 ± 6.18E-03 | 2.047 ± 8.0E-04 | 2.125 ± 4.50E-04 | 2.138 ± 3.30E-04 | 2.139 ± 2.70E-04 | 2.146 ± 3.50E-04 | 2.137 ± 2.80E-04 | 2.154 ± 7.20E-04 |
| Guardian | 2.048 ± 5.45E-03 | 1.944 ± 1.14E-03 | 2.012 ± 5.50E-04 | 2.047 ± 4.50E-04 | 2.057 ± 5.20E-04 | 2.056 ± 6.0E-04 | 2.050 ± 5.70E-04 | 2.068 ± 1.40E-04 |
| Reuters | - | - | 2.402 ± 1.18E-02 | 2.229 ± 8.20E-03 | 2.180 ± 1.087E-02 | 2.136 ± 1.60E-02 | 2.112 ± 4.60E-02 | 2.145 ± 1.76E-02 |

to the $j^{th}$ dictionary is given, for $j = 1, \ldots, p$, by:

$$H_N^{(j)}(\tilde{v}_N) | \{g_N^{j,l}(\cdot)\}_{l=1:q_j} = \begin{cases} -\sum_{l=1}^{q_j} \mathbb{I}^{j,l}(\tilde{v}_N) g_N^{j,l} \ln\left(g_N^{j,l}(\cdot)\right), & \exists\, l \text{ s.t. } g_N^{j,l}(\tilde{v}_N, \mathcal{D}^{i,l}) \neq 0 \\ 0, & \text{otherwise.} \end{cases} \quad (9)$$

Using this framework, we construct the sentiment index per news source and provide the robust median of the sentiment per month during the onset of the pandemic. We demonstrate this descriptive summary statistic in Table 4 where we have also included the 95% confidence intervals.

To finalise this process in order to make it applicable for incorporation as a daily exposure modulation in our proposed regression structures, we next have to make from the document indexes of time which run on an $n$-gram time scale, a time scale commensurate with the observed daily infection counts. Hence, we need to align the time index of the text time-series and the calendar time. This is achieved by combining all the news sources and associated sentiment summaries into a single point estimator for each day.

We have discussed how the dictionary plays a critical role in the construction of the sentiment index. The importance of its content becomes especially relevant for the sentiment time-series, as it will determine what type of sentiment the time-series captures. Our approach allows us to distinguish between two complementary, in terms of impact on policy-making, sentiment types: sentiment related to population health and life sciences, and sentiment related to economic impact. Policymakers need to consider both when deciding on imposing or lifting countermeasures for COVID-19. Depending on the dictionary we employ, we could therefore construct two distinct sentiment indices, one for each sentiment, or alternatively, an index for the global sentiment magnitude that captures both sentiment types. The latter is the one we utilised in our studies.

With these remarks in mind we present the following method to construct text-based sentiment time-series.

Let $Y_{1:N^{s,j}}^{(s,j)}$, $s \in \{\text{health, economics, health} \cup \text{economics}\}$, $j \in \{\text{NYT, ECDC, USCDC, WHO, UNECE, Telegraph, Guardian, Reuters}\}$ be the text time-series corresponding to each source, where $N^{s,j}$ denotes the total number of $n$-grams of source $j$, with sentiment $s$. For calendar time units $t = 1, \ldots, T$ we can segment $Y_{1:N^{s,j}}^{s,j}$ by grouping the observations that come from articles published on the same day: $Y_{1:n_1^{s,j}}^{(s,j)}, \ldots, Y_{1:n_T^{s,j}}^{(s,j)}$, where $n_t^{s,j} \geq 0$. Note that with respect to sentiment, we may have different sentiment indices according to the dictionary we have used in the construction, as we noted earlier.

We combine the sentiment time-series of the different news sources together, using as weight for each daily observation the number of $n$-grams generated from the corresponding news source. The combined sentiment index for all sources is then:

$$Z_t^s = \sum_j w_t^{s,j} \tilde{Y}_t^{(s,j)}, \ \ w_t^{s,j} = \frac{n_t^{s,j}}{\sum_j n_t^{s,j}}, \tag{10}$$

where $\tilde{Y}_t^{(s,j)}$ denotes the daily summary of each partition $Y_{1:n_t^{s,j}}^{(s,j)}$ of source $j$ and sentiment $s$ that corresponds to time $t$. The daily summary used in this work is the interquartile range, which captures the volatility of the news reporting regarding COVID-19. The weights are assigned according to the volume of $n$-grams per day for each source, which ensures that article lengths have no effect on the weight. Please refer to the S1 Appendix for the algorithm describing the daily sentiment index construction (Section B in S1 Appendix).

## 5 Bayesian model estimation framework via `RStan`

In this section we detail the model estimation and assessment framework for the models of Eqs 1 and 2 using the total number of infected cases for the UK, Australia, Germany, Italy, Spain, US and Japan. We study the in-sample model fitting, out-of-sample forecasting, and conduct a comparison study of in-sample modelling with and without sentiment data $E_t$ utilised as an exposure adjustment. In the in-sample modelling study, a Bayesian approach is adopted to compare the feasibility of various latent processes. The model risk, especially in terms of prediction, is revealed by the out-of-sample forecasting study. We will demonstrate in the results section that by incorporating the sentiment information $E_t$, the in-sample model performance can be significantly improved and the model risk can be reduced especially in the early stages of the disease spread where there is significant fear, uncertainty and doubt present that we conjecture leads to many people paying particular attention to official health news and announcements before saturation of such news took effect.

To implement the proposed models efficiently, as already mentioned we chose the Bayesian R package `Rstan` which utilises the `STAN` program within `R` developed in the `C++` language. The Hamiltonian Monte Carlo (HMC) sampler [31, 32] is an extension of the class of MCMC sampling methods that is adopted in `Rstan`. For complex Bayesian posterior models with many parameters such as the models developed in this manuscript, the HMC sampler converges faster than the conventional samplers such as random-walk Metropolis and Gibbs sampler.

In order to closely monitor the dependence, precision, and convergence of posterior samples, three measures are reported in `Rstan`. The first measure is the *number of effective samples* which indicates the effective posterior sample size after allowing for the dependence within a Monte Carlo sample. The second measure is the *Monte Carlo standard error* (MCSE)

$$\text{MCSE} = \frac{\text{posterior standard deviation}}{\sqrt{\text{number of effective samples}}},$$

which reports the error of estimation for the posterior mean. We carefully monitored the convergence behaviour of all Markov chain Monte Carlo solutions from the HMC sampler via standard convergence diagnostic measures for $k > 2$ multiple runs of Markov chains of length $2n$ each. We monitored the convergence via measures such as those proposed in the work of [33], given by the Gelman-Rubin $\hat{R}$ statistic and the Geweke Z-score as well as the effective sample size; details are provided in the online supplementary appendix (Section A in S1 Appendix) as such measures are standard in `RStan` package. In the studies performed, the

number of chains was $k = 10$ and for each chain there was overall a total of $n = 100,000$ Markov chain iterations, with the first $10,000$ iterations discarded as burn-in. Hence, there are $L = 90,000$ subsequent iterations with thin set to 1. The values of $\hat{R}$ for each estimator and the history plot are carefully checked to ensure that all parameters meet the convergence condition. For all of in-sample fitting and out-of-sample forecast studies, the number of effective samples ranges from 75,000 to 86,000 across all model parameters in the posterior and for all chains. The range of $\hat{R}$ is between 1.0000 and 1.0003, which indicates moderate dependency and clear convergence.

## 5.1 Bayesian model selection and forecast performance

The performance of each model is evaluated through a popular Bayesian model selection criterion called deviance information criterion (DIC) [34]. As a generalisation of Akaike's Information Criterion (AIC), DIC can deal with models containing informative priors, such as hierarchical models. As the priors can effectively restrict the freedom of model parameters, the number of parameters as required in the calculation of AIC is generally unclear. DIC overcomes such problems by providing an estimate for the effective number of parameters. The DIC can be calculated using the equation:

$$
\begin{aligned}
\text{DIC} \quad &= \bar{D} + p_D = 2\bar{D} - D(\bar{\boldsymbol{\vartheta}}_x) \\
&= 2\mathbb{E}_{\boldsymbol{\vartheta}_x|y_x}[-2\ln{(f(\boldsymbol{y}_x|\boldsymbol{\vartheta}_x))}] - (-2\ln{(f(\boldsymbol{y}_x|\boldsymbol{\vartheta}_x))}),
\end{aligned}
\tag{11}
$$

where $D(\boldsymbol{\vartheta}_x) = -2\ln{(f(\boldsymbol{y}_x|\boldsymbol{\vartheta}_x))}$ is the deviance, $\bar{D} = \mathbb{E}_{\boldsymbol{\vartheta}_x|y_x}[-2\ln{(f(\boldsymbol{y}_x|\boldsymbol{\vartheta}_x))}]$ measures the model fit, $p_D = \bar{D} - D(\bar{\boldsymbol{\vartheta}}_x)$ is the estimated number of parameters and measures model complexity, and $f(\boldsymbol{y}_x|\boldsymbol{\vartheta}_x)$ is the likelihood function, namely Eq 3 in this case.

Considering the $m$-step ahead forecasts $\hat{y}_{x,t}$ given by the posterior mean or median and the observations $y_{x,t}$ with $T$ time points and $g$ groups, e.g. age groups, the forecast performance can be evaluated by adopting three types of measures, namely residuals $r_{x,t} = y_{x,t} - \hat{y}_{x,t}$, percentage errors $p_{x,t} = \frac{r_{x,t}}{y_{x,t}} \times 100\%$ and scaled errors $\epsilon_{x,t}$ defined in Eq 14.

Based on $r_{x,t}$ and $p_{x,t}$, three popular criteria, namely mean absolute error (MAE), root mean squared error (RMSE) and mean absolute percentage error (MAPE), are defined respectively below

$$
\text{MAE} = \frac{1}{g}\sum_{x=1}^{g}\left[\frac{1}{m}\sum_{t=1}^{m}|r_{x,T+t}|\right], \quad \text{RMSE} = \sqrt{\frac{1}{g}\sum_{x=1}^{g}\left[\frac{1}{m}\sum_{t=1}^{m}r_{x,T+t}^2\right]},
$$
$$
\text{and} \quad \text{MAPE} = \frac{1}{g}\sum_{x=1}^{g}\left[\frac{1}{m}\sum_{t=1}^{m}|p_{x,T+t}|\right].
\tag{12}
$$

However, $r_{x,t}$ are scale-dependent making comparison difficult, and although $p_{x,t}$ are scale-free, they are sensitive to observations close to zero. Hence the fourth criterion we adopt is the mean absolute scaled error (MASE) defined as

$$
\text{MASE} = \frac{1}{g}\sum_{x=1}^{g}\left[\frac{1}{m}\sum_{t=1}^{m}|\epsilon_{x,T+t}|\right],
\tag{13}
$$

making use of the scaled errors

$$\epsilon_{x,T+t} = \frac{r_{x,T+t}}{\frac{1}{m-1}\sum_{t=2}^{m}|y_{x,T+t} - y_{x,T+t-1}|}, \tag{14}$$

proposed by [35].

A similar approach can also be applied to evaluate estimated results $\hat{\mu}_{x,t}$ calculated by the posterior mean or median. Hence, the residuals $r_{x,t}^s = \mu_{x,t} - \hat{\mu}_{x,t}$, percentage errors $p_{x,t}^s = \frac{r_{x,t}^s}{\mu_{x,t}} \times 100$ and scaled errors

$$\epsilon_{x,t}^s = \frac{r_{x,t}^s}{\frac{1}{m-1}\sum_{t=2}^{m}|\mu_{x,t} - \mu_{x,t-1}|},$$

can be used to construct similar criteria for the $\mu$ estimator, namely the MAE, RMSE, MAPE and MASE by using the same formulas in Eqs 12 and 13.

In order to measure the forecast performance of the models in Eqs 1 and 2, the numbers of infected counts $Y_{1:T}$ are divided into two parts, the training $Y_{1:(T-20)}$ and forecast $Y_{(T-19):T}$.

## 6 Results

We have separated the results and subsequent analysis into two stages, i.e. pre- and post-vaccine, as these two periods corresponded to different phases of epidemic behaviour, policy and news reporting. The pre-vaccine phase corresponds to the period from around January 2020 to around the start of August 2020. The second component of analysis will analyse data from January 2020 through to January 2021 and will therefore incorporate the pandemic's behaviour post the roll-out of vaccination programs in many countries. We include the results for the UK, Germany, the US and Australia in the following sections, and the results for Spain, Italy and Japan are included in the (Sections C—F in S1 Appendix).

### 6.1 Analysis covering the pre-vaccination phase: January 2020—Early August 2020

In this study, the observation function is modelled by a two-component spliced distribution (see Definition 2.1) as this was deemed appropriate for the pre-vaccine phase. This is due to the fact that in this time period one clear wave of epidemic had occurred and the second wave was just starting to initiate. We found that this period would require distinction in the cumulative cases of early phase with no community spreading vs wide spread community transmission in the first wave of infection. This distinction was accommodated by introducing the splicing in our modelling determined by the threshold $y_*$.

**6.1.1 Selection of the splice threshold.** To assess the impact of the choice of threshold $y_*$, the M2 model, namely the stochastic Gompertz reference model, is fit to each of the datasets. We illustrate the results in the case of the German data in Table 5. According to the DIC results reported, the choice of threshold $y_*$ makes a difference in model performance and the optimal choice is obtained when $y_* = 200$. The results for a two component splice model were

**Table 5. DIC (Eq 11) of stochastic Gompertz model (M2) fit to data for Germany for various splice thresholds.**

| threshold | 50 | 100 | 150 | 200 | 250 | 300 | 400 | 500 |
|---|---|---|---|---|---|---|---|---|
| DIC | 2851.357 | 2847.432 | 2839.313 | *2838.173* | 2838.772 | 2841.038 | 2848.024 | 2852.487 |

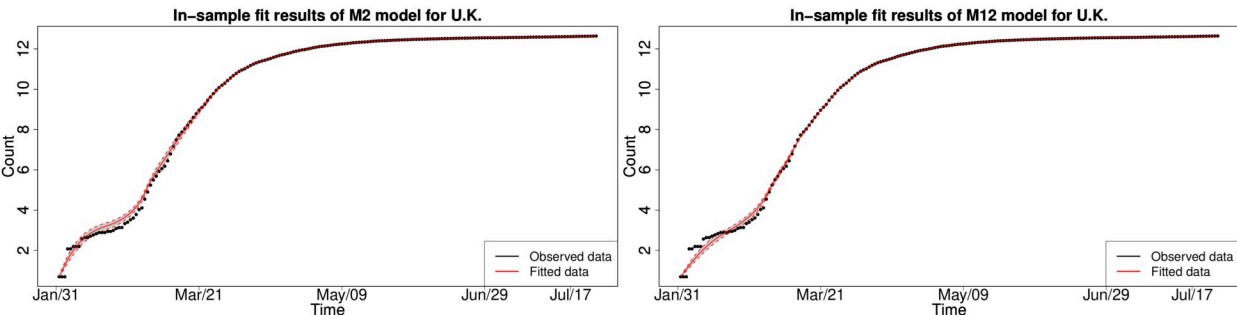

**Fig 6. In-sample fit results for the UK for the period January 2020—August 2020.** In-sample fitted plot (y-axis in log scale) for the UK by Model 2 (baseline, left) and Model 12 (best, right) (January 2020—August 2020).

comparable across the other countries when this study was undertaken and, consequently, the value of $y_*$ will be fixed to 200 in the following studies for the pre-vaccine phase of analysis.

**6.1.2 Bayesian in-sample growth model calibration analysis.** We consider the in-sample fit results obtained from the `Rstan` HMC Markov chain samples from each of the Bayesian splice models with the selected splice threshold $y_* = 200$. We found that not all models are able to adequately capture the dynamics of the national-level cumulative COVID-19 infections for each country's national cumulative daily infections. Therefore, we have selected to focus on the subset of models that fit all countries in a reasonable fashion after convergence analysis of the HMC Markov chains. The chains were assessed as statistically converged according to Effective Sample size and other standard MCMC convergence diagnostics such as Geweke Z-score and Gelman-Rubin statistics. As a consequence, we were left to consider a subset of the stochastic growth trend models M2, M4, M7, M8, M9, M10, M11 and M12 for the period of January 2020 to August 2020.

**6.1.3 In-sample fitting results for the UK, Germany, the US and Australia.** For the period from January 2020 to August 2020, Figs 6–9 show the time-series plots after applying a log transform (black trace), against the estimated in-sample fitting results (red trace) with credible intervals (grey band). The model performance is compared among M2 (baseline Gompertz model) and Models 7 or 12, which were the best fitting models.

In Figs 6 and 7 it is observed that M12 and M7, as best models, can better capture the data characteristics and tendency for the UK and Germany, hence provide a meaningful guidance for policymakers. In addition, Fig 8 illustrates that the gap in model fitting between the best model and the baseline model can be significant for US data, which may cause misleading

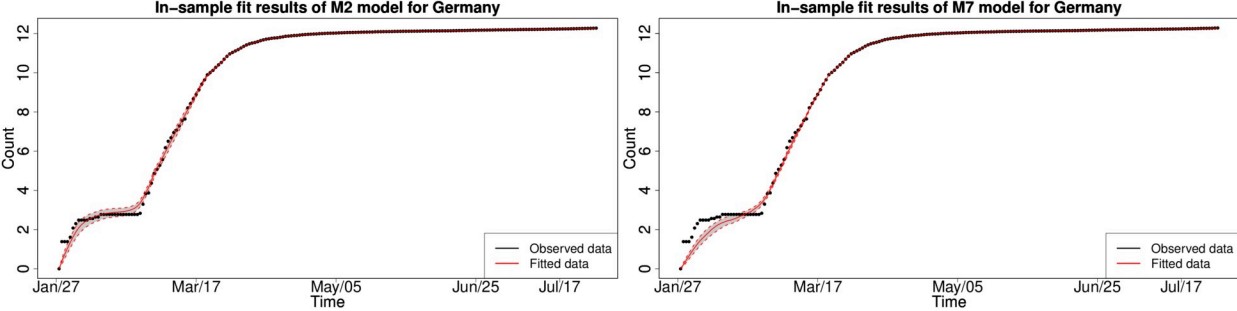

**Fig 7. In-sample fit results for Germany for the period January 2020—August 2020.** In-sample fitted plot (y-axis in log scale) for Germany by Model 2 (baseline, left) and Model 7 (best, right) (January 2020—August 2020).

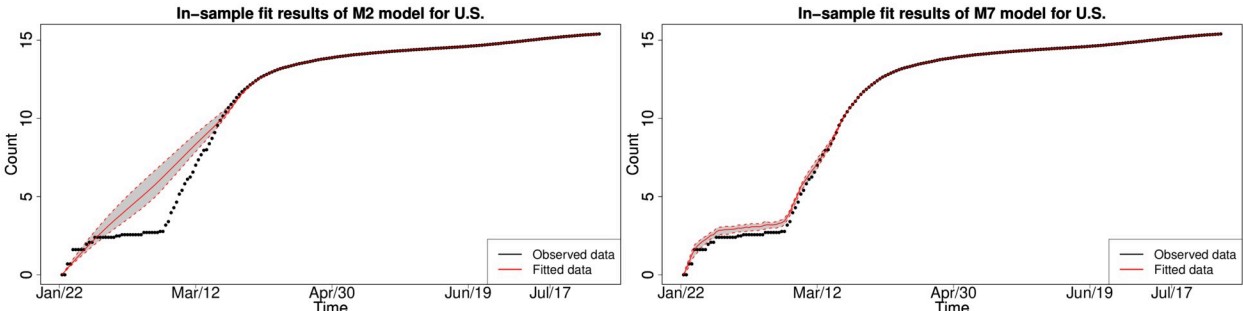

**Fig 8. In-sample fit results for the US for the period January 2020—August 2020.** In-sample fitted plot (y-axis in log scale) for the US by Model 2 (baseline, left) and Model 7 (best, right) (January 2020—August 2020).

information in making epidemic prevention measures. Finally, Fig 9 shows that M2 provides better model fitting for Australia time-series with narrower credible interval width, which means the accuracy of the best fitting model is higher than the other models.

Evidence for these qualitative observations is demonstrated in Table 6 which shows the model performance of the considered different models when fit to the seven countries of the study. The best models with the smallest DIC values are shaded in grey. For most time-series, M7 and M12 outperform other models with smaller DIC values. M2 as a baseline model is slightly better than M7 and M12 for Japan and Australia. For the US time-series, the trend can only be properly fit by M7, M9 and M12, with M7 being the best choice.

## 6.2 Analysis covering pre- and post-vaccination phases: January 2020— January 2021

In this section we extend the previous study to include the pre-vaccine first wave of the COVID-19 epidemic as well as the second wave of the epidemic that occurred during the onset of vaccination programs. This second set of studies therefore covered a period of analysis

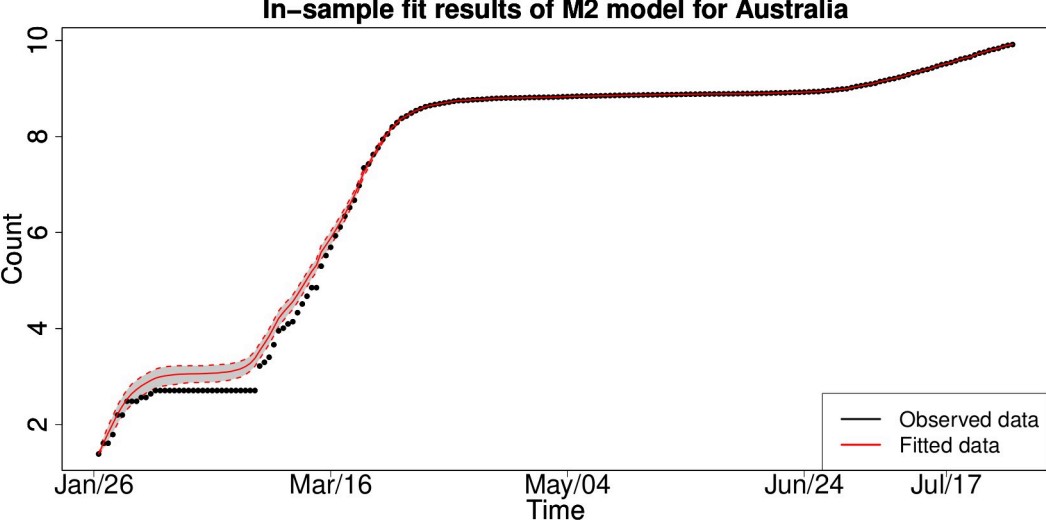

**Fig 9. In-sample fit results for Australia for the period January 2020—August 2020.** In-sample fitted plot (y-axis in log scale) for Australia by Model 2 (best) (January 2020—August 2020).

**Table 6. DIC (Eq 11) for fitted models in the pre-vaccine period (January 2020—August 2020).**

| Data type | M2 | M4 | M7 | M8 | M9 | M10 | M11 | M12 |
|---|---|---|---|---|---|---|---|---|
| UK | 2.80E+03 | 3.72E+03 | 2.21E+03 | 4.17E+04 | 2.79E+03 | 2.76E+03 | 2.73E+03 | *2.20E+03* |
| SP | 2.73E+03 | 5.66E+03 | 2.53E+03 | 8.91E+04 | 2.73E+03 | 2.63E+03 | 2.72E+03 | *2.51E+03* |
| IT | 2.93E+03 | 3.84E+03 | *2.27E+03* | 3.03E+04 | 2.91E+03 | 2.88E+03 | 2.89E+03 | 2.34E+03 |
| GM | 2.84E+03 | 4.19E+03 | *2.41E+03* | 9.34E+04 | 2.81E+03 | 2.66E+03 | 2.68E+03 | 2.42E+03 |
| US | 2.97E+04 | 1.56E+07 | *2.77E+03* | 2.23E+05 | 3.20E+03 | 2.10E+04 | 7.58E+04 | 2.83E+03 |
| JP | *2.17E+03* | 2.82E+03 | 2.21E+03 | 5.91E+03 | 2.18E+03 | 2.19E+03 | 2.19E+03 | 2.20E+03 |
| AU | *2.04E+03* | 2.49E+03 | 2.05E+03 | 4.84E+03 | 2.04E+03 | 2.05E+03 | 2.04E+03 | 2.05E+03 |

from January 2020 through to January 2021 in which there are clearly three phases of the epidemic, the early local isolated epidemic events in January through to March, followed by wave one of the epidemic from March through to May-June, punctuated by a latency period before the onset of wave two of the epidemic as Winter in the Northern hemisphere began. The second wave was characterised by a very rapid increase in the daily cumulative trend, even faster than the growth rates in the first wave of the epidemic. Obtaining models that characterise such structures is challenging and so we considered to extend the original splice model to a splice model with two splice breaks $y_{*,1}$ and $y_{*,2}$. We kept $y_{*,1} = 200$ from our pre-vaccine phase analysis and then compared the M2 stochastic Gompertz model with one or two splice levels to see if a second splice phase was warranted. The details of this aspect are presented in the following section.

**6.2.1 Bayesian in-sample growth model calibration analysis.** We begin by presenting the analysis for the model with a single splice threshold $y_* = 200$ as studied previously, now applied to the entire pre- and post-vaccine periods. Table 7 lists DIC values calculated by M2 (baseline model), M7 and M12 (best models). For the full-length dataset, M12 shows better performance than M2 and M7 in all countries except Australia, even though the scores for the two models seem to be very close.

**6.2.2 In-sample fitting results for the UK, Germany, and Australia.** To compare the model performance for the different data lengths, we look at Figs 6–9 (January 2020—August 2020), and Figs 10–14 (January 2020—January 2021). We observe that M2 is not suitable for the data that span both the pre- and the post-vaccine phases, which means that the flexibility of M2 is not high enough to allow steep changes of the cumulative infected counts. M12 outperforms the rest of the models with narrower credible intervals for the UK fit. For Australia with three distinct stages in the evolution of the spread (e.g. see the left panel in Fig 14), M7 can easily fit this type of time-series. Overall, both M7 and M12 are able to capture the steep changes, and the model performance of M12 is slightly better than M7.

**6.2.3 In-sample fitting results for the US.** A notable exception in this fitting process is the US, where the second wave of COVID-19 was more pronounced in comparison to the rest of the countries analysed. It was determined that as a consequence, for any of the models to adequately capture this extreme second wave of infections, it would be required to add a second splice threshold level.

**Table 7. DIC (Eq 11) for Model 2, Model 7 and Model 12 in the pre- and post-vaccine period (January 2020—January 2021).**

| Data type | UK | SP | IT | GM | US | JP | AU |
|---|---|---|---|---|---|---|---|
| M2 | 8.26E+03 | 1.13E+04 | 7.78E+03 | 6.32E+03 | 9.66E+04 | 5.60E+03 | 5.10E+03 |
| M7 | 5.80E+03 | 5.77E+03 | 6.03E+03 | 5.76E+03 | 9.31E+04 | 4.38E+03 | *4.239E+03* |
| M12 | *5.76E+03* | *5.72E+03* | *5.97E+03* | *5.70E+03* | *8.24E+03* | *4.36E+03* | 4.24E+03 |

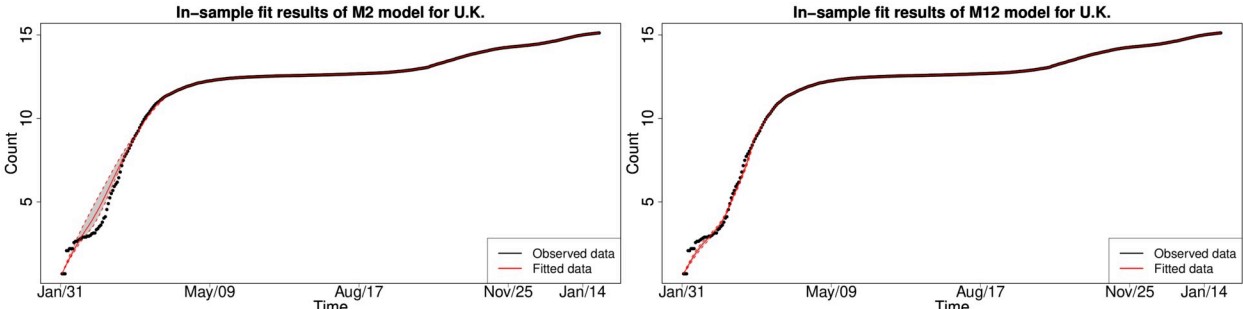

**Fig 10. In-sample fit results for the UK for the period January 2020—January 2021.** In-sample fitted plot (y-axis in log scale) for the UK by Model 2 (baseline, left) and Model 12 (best, right) (January 2020—January 2021).

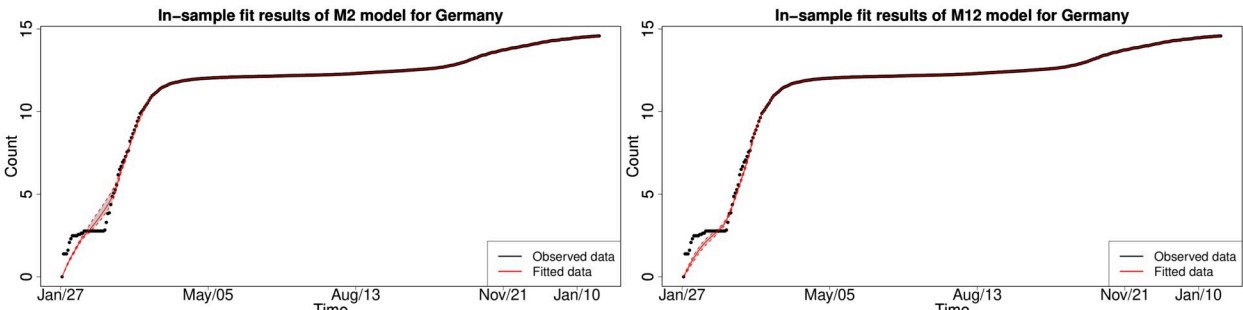

**Fig 11. In-sample fit results for Germany for the period January 2020—January 2021.** In-sample fitted plot (y-axis in log scale) for Germany by Model 2 (baseline, left) and Model 12 (best, right) (January 2020—January 2021).

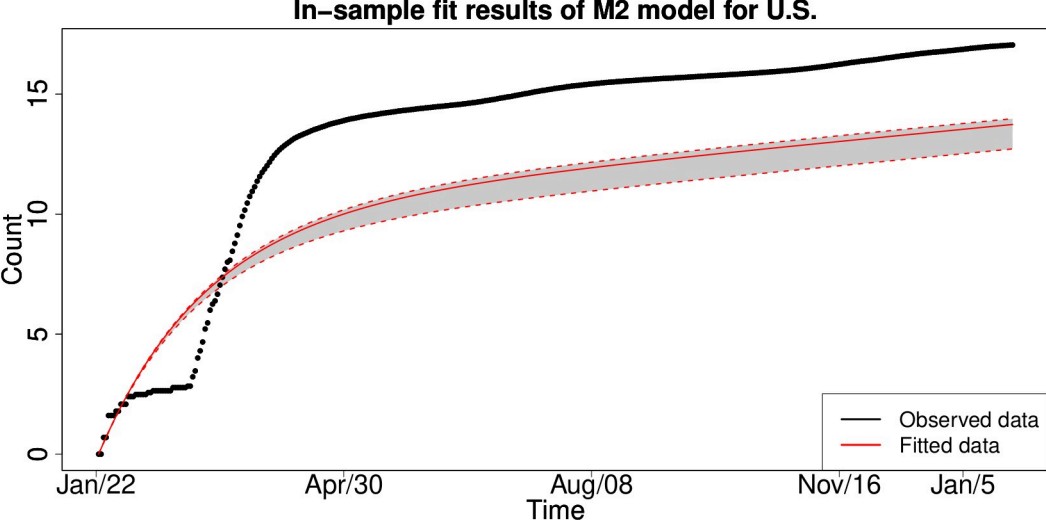

**Fig 12. In-sample fit results for the US for the period January 2020—January 2021.** In-sample fitted plot (y-axis in log scale) for the US by Model 2 (baseline) (January 2020—January 2021).

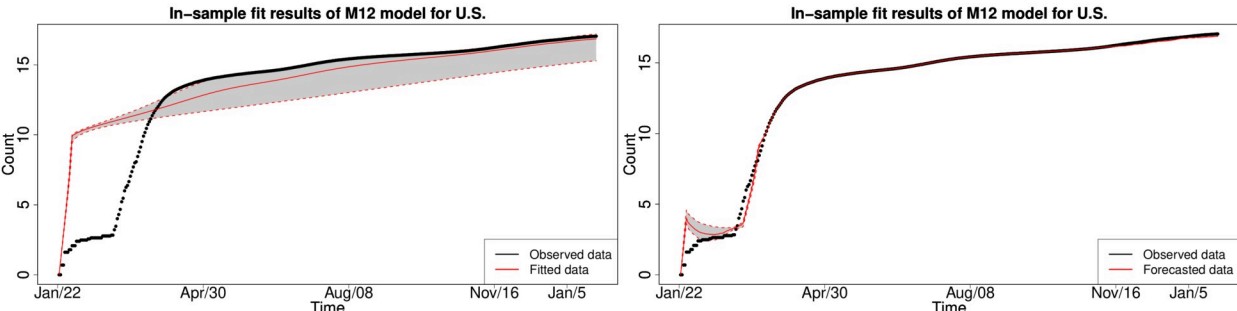

**Fig 13. In-sample fit results for the US for the period January 2020—January 2021.** In-sample fitted plot (y-axis in log scale) for the US by Model 12 single splice (left) and Model 12 2-splice (right) (January 2020—January 2021).

We kept $y_{*,1} = 200$ and we incorporated a second splice threshold $y_{*,2}$ to accommodate the distinct rate of growth of the US infected counts that caused a steeper increment in the infected counts than the rest of the countries. We performed a grid search on threshold values evaluating the DIC to select the second splice threshold, starting from the first threshold which is common to all models. The final value for the second splice threshold was eventually set to $y_{*,2} = 10,000,000$ which corresponded to a time period around the beginning of November 2020, which naturally coincides with the onset of the second epidemic wave at early stages of Winter in the Northern hemisphere. The difference in the fitting was significant as we can see in Fig 13 where we compare the models with one (left) vs with two (right) splice thresholds. The model with two thresholds (whose DIC we report in Table 7) is clearly able to fit the data in both phases of the infection spread in the US, as opposed to both M12 with just a single splice, and the baseline M2 (Fig 12).

## 6.3 Out-of-sample forecast study

In this section, according to the in-sample fitting results, M2 (baseline), M7 (best model), and M12 (best model) are selected to evaluate the model performance in out-of-sample forecasting with models fitted for data in the period from January 2020 to January 2021. We calculate the 20-step ahead forecast based on the posterior predictive distributions and the posterior sample size of $L = 90,000$. Table 8 reports the four forecast performance criteria of Section 5.1 for the four models. The four criteria are calculated based on posterior predictive mean estimators and forecast set $Y_{(T-19):T}$. The minimum values are shaded by grey, which represent the best model performance in out-of-sample forecasting.

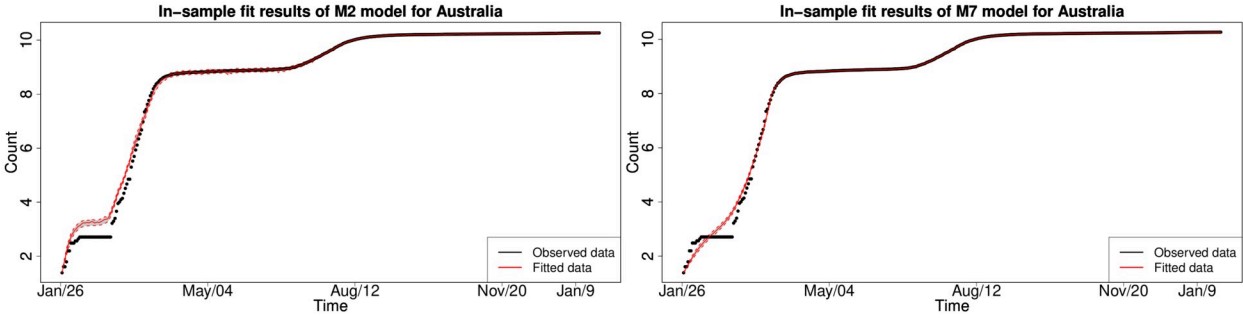

**Fig 14. In-sample fit results for Australia for the period January 2020—January 2021.** In-sample fitted plot (y-axis in log scale) for Australia by Model 2 (baseline, left) and Model 7 (best, right) (January 2020—January 2021).

**Table 8. Forecasting criteria results using baseline and best fitting models (fitted on data from January 2020—January 2021).**

| Data type | UK | SP | IT | GM | US | JP | AU |
|---|---|---|---|---|---|---|---|
| | | | | MAE | | | |
| M2 | *1.77E+05* | 2.73E+05 | 1.14E+05 | *8.40E+04* | 1.35E+06 | *2.86E+04* | 4.00E+02 |
| M7 | 2.03E+05 | 2.70E+05 | 1.12E+05 | 9.61E+04 | 1.36E+06 | 2.93E+04 | *1.17E+01* |
| M12 | 1.97E+05 | *2.65E+05* | *1.07E+05* | 9.03E+04 | *0.93E+06* | 2.90E+04 | 8.43E+01 |
| | | | | RMSE | | | |
| M2 | *1.94E+05* | 3.01E+05 | 1.30E+05 | *9.19E+04* | 1.49E+06 | *3.11E+04* | 4.69E+02 |
| M7 | 2.24E+05 | 2.97E+05 | 1.27E+05 | 1.06E+05 | 1.51E+06 | 3.19E+04 | *1.24E+01* |
| M12 | 2.17E+05 | *2.90E+05* | *1.21E+05* | 9.92E+04 | *1.19E+06* | 3.16E+04 | 9.98E+01 |
| | | | | MAPE | | | |
| M2 | *4.50E-02* | 9.28E-02 | 4.33E-02 | *3.67E-02* | 4.99E-02 | *7.09E-02* | 1.39E-02 |
| M7 | 5.15E-02 | 9.17E-02 | 4.24E-02 | 4.20E-02 | 5.04E-02 | 7.25E-02 | *4.05E-04* |
| M12 | 5.01E-02 | *8.98E-02* | *4.04E-02* | 3.95E-02 | *3.43E-02* | 7.19E-02 | 2.92E-03 |
| | | | | MASE | | | |
| M2 | *9.65E+00* | 1.22E+01 | 9.20E+00 | *8.98E+00* | 1.17E+01 | *1.25E+01* | 6.67E+01 |
| M7 | 1.10E+01 | 1.20E+01 | 9.02E+00 | 1.03E+01 | 1.18E+01 | 1.27E+01 | *1.95E+00* |
| M12 | 1.08E+01 | *1.18E+01* | *8.59E+00* | 9.66E+00 | *0.81E+01* | 1.26E+01 | 1.40E+01 |

**6.3.1 Out-of-sample forecast results for the UK, Germany, and Australia.** For the UK (Fig 15) and Germany (Fig 16), the model performance of M2 approaches generally the same level of accuracy as M7 and M12 in the forecast performance, despite the fact that in-sample fits of M2 for the full range of data are not as good as those obtained for M7 and M12 in-sample as demonstrated in Table 7. We note that the in-sample fit superior Model 12 does provide slightly narrower credible intervals than M2 for Germany, as we see in Fig 16, contrary to the UK, where the credible intervals appear slightly narrower for M2 (Fig 15).

Finally, regarding Australia (Fig 17), M7 is distinctly superior than the baseline, both in terms of point estimates and credible intervals.

**6.3.2 Out-of-sample forecast results for the US.** For the US, the model predictability of M12 is distinctly superior to other models in terms of the obtained point estimates, yet it provides wider credible intervals compared to the baseline (Fig 18). Note that in this case, the baseline Model 2 (Fig 18, left) fails to adequately produce forecasts that were reliable during this period, hence the lack of credible intervals in the figure. The Markov chain was mixing adequately in-sample but the forecast posterior predictive intervals were unreliable. Overall, our conclusion for the stochastic Gompertz model M2 in the US case of pre- and post-vaccine

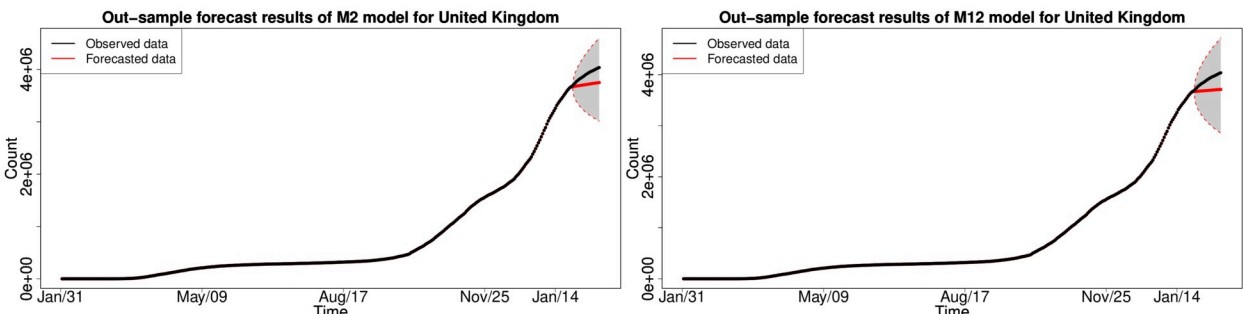

**Fig 15. Out-of-sample forecast results for the United Kingdom.** Out-of-sample forecast plot for the UK by Model 2 (left) and Model 12 (right).

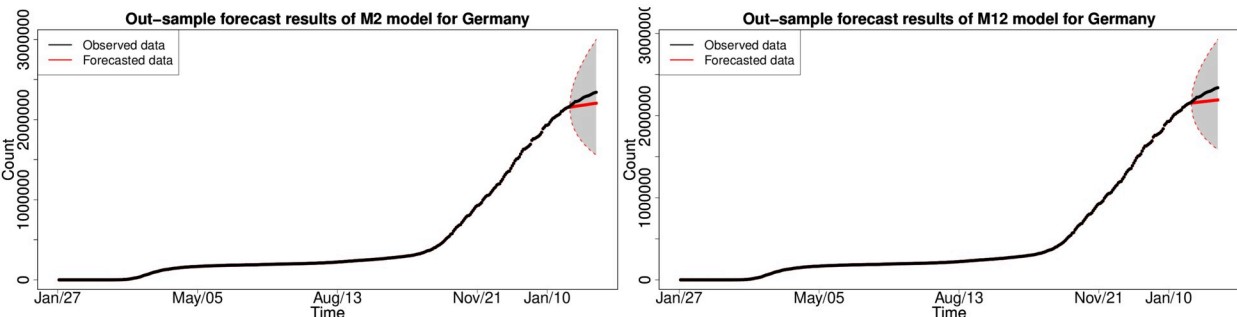

**Fig 16. Out-of-sample forecast results for Germany.** Out-of-sample forecast plot for Germany by Model 2 (left) and Model 12 (right).

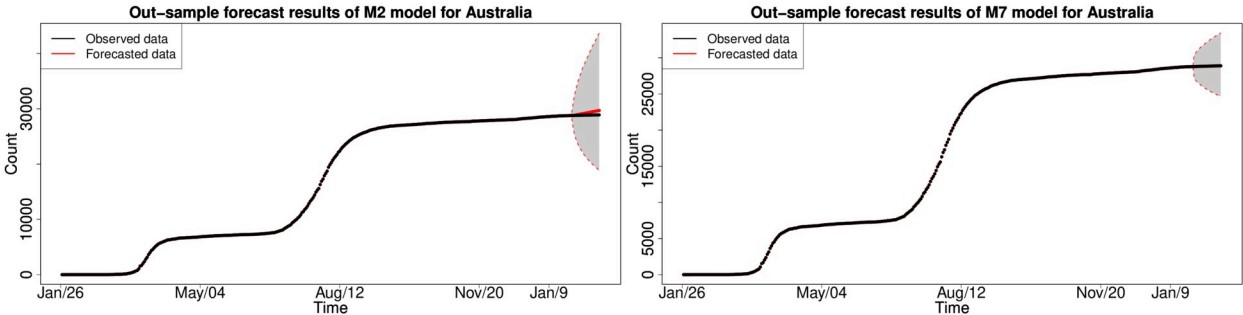

**Fig 17. Out-of-sample forecast results for Australia.** Out-of-sample forecast plot for the Australia by Model 2 (left) and Model 7 (right).

phases was that it failed to provide adequate fit performance. We checked that this was not a result of the HMC sampler performance but rather a failure in the model flexibility.

## 6.4 News sentiment exposure-adjusted stochastic observation models study

In this section we extend the models developed in the previous analysis to include the introduction of news sentiment. The sentiment indices were purpose-built in order to capture the informational content and perception in the public of health reporting and news reporting on COVID-19. They were extracted via natural language processing techniques detailed in Section 4.2. The sentiment index extracted was then combined into the stochastic growth models

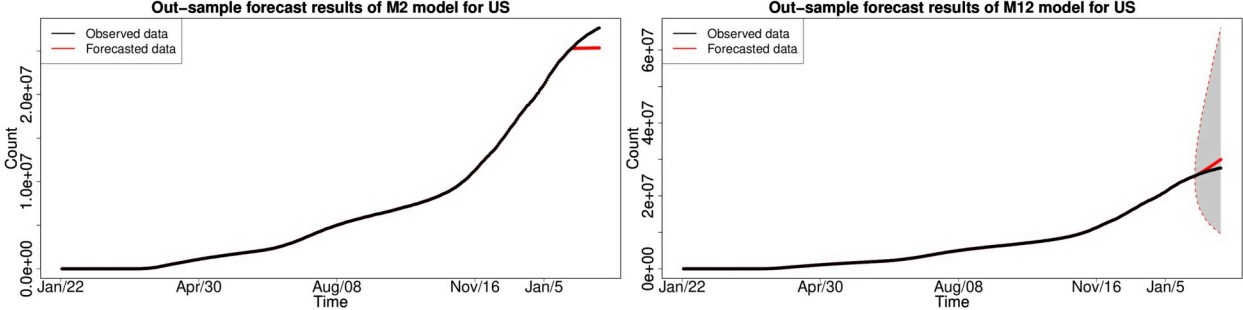

**Fig 18. Out-of-sample forecast results for the United States.** Out-of-sample forecast plot for the US by Model 2 (left) and Model 12 (right).

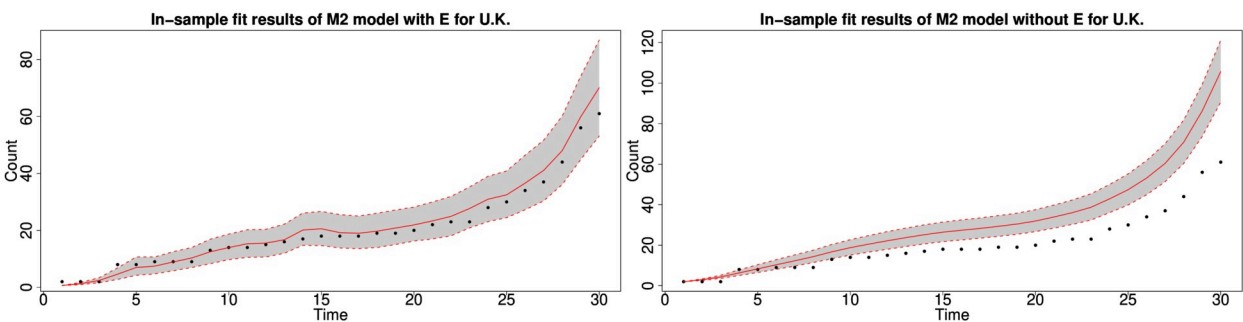

**Fig 19. In-sample fit results with the sentiment exposure adjustment for the United Kingdom.** In-sample fitting plot for the UK by Model 2 for the first month with (left) and without (right) the sentiment exposure adjustment.

via an exposure adjustment of the linear predictor of each model. We seek to quantify and verify if there is a measurable effect of public health reporting and COVID-19 news reporting on people's behaviour as quantified by changes in the growth rates of national daily cumulative infections. Presumably, if a public health policy is being effectively communicated and adhered to, then such measures will reduce over time the potential for community spread, thereby resulting in reduced daily infection rates.

Therefore, in this section we focus on the natural language exposure adjustment and show how it affects the baseline model by improving the in-sample fit, which may be valuable in model assessment and consequently in reducing the risk associated with selecting an appropriate model. We only investigated the baseline M2 in this study to explicitly measure the improvements contributed by the sentiment covariate, as opposed to having to disambiguate between sentiment contributions and contributions to performance by a more flexible model.

As we remarked in Section 2.3, we can extend the exposure adjustment $E$ to the link function to be a continuous adjustment function $\tilde{E}_t$, through which we incorporate the natural language sentiment covariate $E_t$ to the observation model. There are several ways to achieve this, such as using a step function, a sigmoid function $\tilde{E}_t = \frac{1}{1+e^{-E_t}}$, or the hyperbolic tangent (tanh) function $\tilde{E}_t = \frac{e^{E_t}-e^{-E_t}}{e^{E_t}+e^{-E_t}}$. In this study, we only adopt the sigmoid function $\tilde{E}_t$ and remark that the values are very similar to the tanh function, which we also experimented with. To assess the feasibility of $\tilde{E}_t$ for modelling, various settings of data length are tested for the sentiment covariate. In our experiments, the data length $T$ is set to $T \in \{49, 56, \ldots, 189, 196\}$.

**6.4.1 In-sample fitting results with the sentiment exposure adjustment for the UK, Germany, Australia and the US.** Figs 19–22 show the improvements contributed by

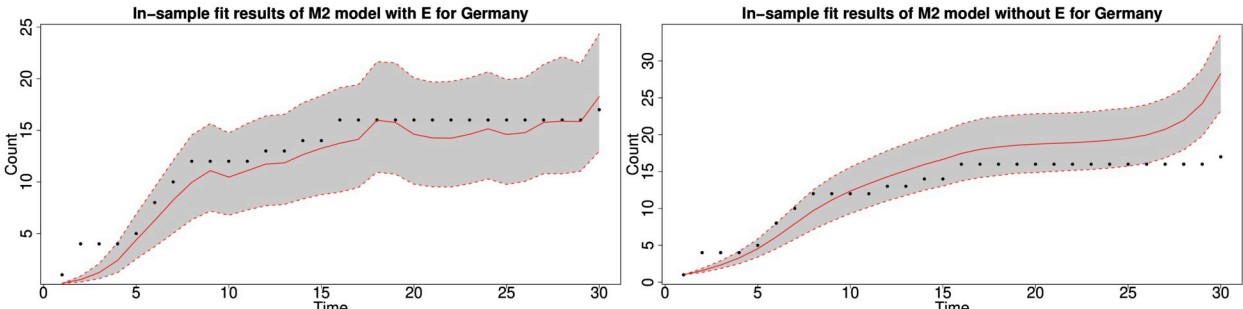

**Fig 20. In-sample fit results with the sentiment exposure adjustment for Germany.** In-sample fitting plot for Germany by Model 2 for the first month with (left) and without (right) the sentiment exposure adjustment.

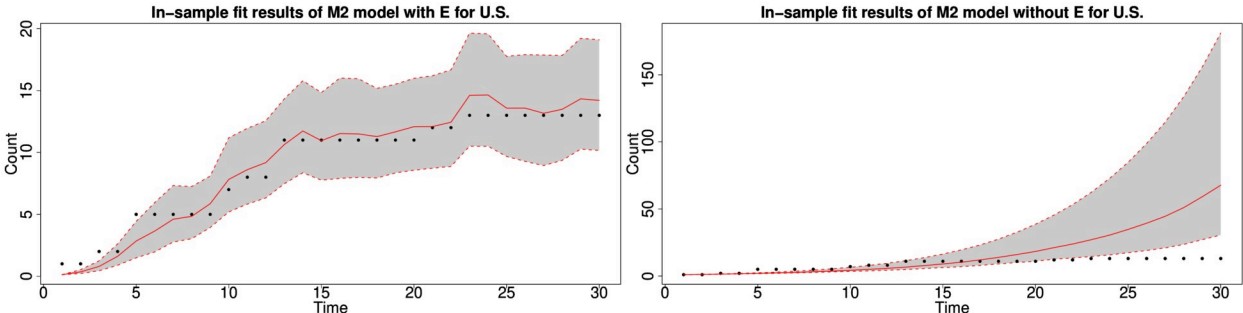

**Fig 21. In-sample fit results with the sentiment exposure adjustment for the United States.** In-sample fitting plot for the US by Model 2 for the first month with (left) and without (right) the sentiment exposure adjustment.

incorporating the sentiment signal exposure adjustments through exposure index $E_t$ in the modelling process using the baseline Model 2. The left panels represent the results obtained by incorporating $E_t$ and the right panels are in-sample fitting results estimated without $E_t$ for the same time period. To demonstrate the significant enhancement by introducing sentiment data $E_t$, the in-sample fit plots for the first month of the pandemic are provided.

We found that including a sentiment index significantly enhanced the model fit and the effect was most pronounced during the early stages of the COVID-19 pandemic. For our modelling, this translates to the better in-sample point estimates obtained by the model that includes the sentiment adjustment. This is evident in the in-sample trace plots, where we see that the mean of the estimated values (red line) is much closer to the true data (black dots) for the models that include the sentiment covariate. In addition, to further illustrate the improvement to the in-sample fit due to the sentiment exposure adjustment, we present the root mean squared error (RMSE) for the data fits of the Figs 19–22 in Table 9. As we observe in the table, the RMSE when we include the sentiment adjustment in the model is significantly lower, up to eight times less, compared to the RMSE for the model without the sentiment exposure adjustment.

## 7 Discussion

### 7.1 Pre-vaccination phase: January 2020—Early August 2020

The results we obtained from this part of the study are interesting in that they demonstrate a clear delineation of performance between the limited flexibility of models M3, M5 and M6 to

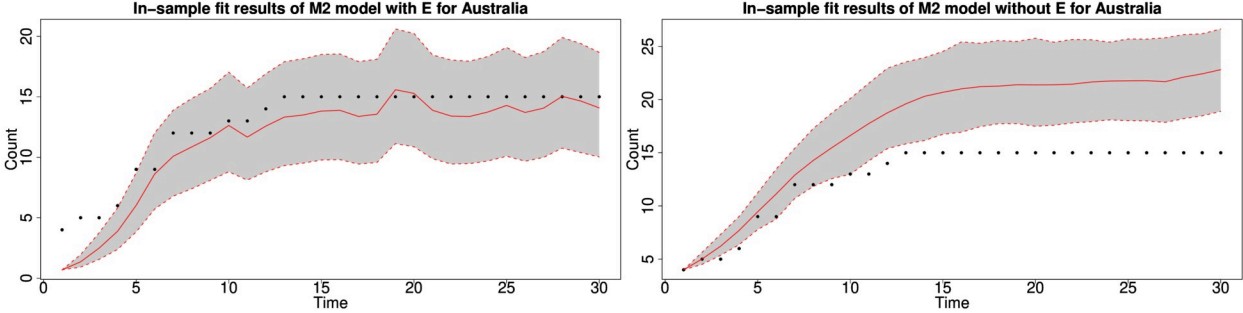

**Fig 22. In-sample fit results with the sentiment exposure adjustment for Australia.** In-sample fitting plot for Australia by Model 2 for the first month with (left) and without (right) the sentiment exposure adjustment.

**Table 9. Root mean squared error (RMSE) for the in-sample fits for model M2 at the beginning of the pandemic, with and without the sentiment exposure adjustment.**

| Country | In-sample fit with sentiment exposure (RMSE) | In-sample fit without sentiment exposure (RMSE) |
|---|---|---|
| US | 0.903 | 7.262 |
| Germany | 1.489 | 5.221 |
| Australia | 1.593 | 7.366 |
| United Kingdom | 2.942 | 17.184 |

capture the characteristics of the national epidemic data for the countries, versus the more flexible models introduced in this manuscript, namely Model 7 through to Model 12. It was quite informative that even after including a splice model structure to provide flexibility in the pre-community spreading and the wide spread community transmission phases of the epidemic, even with this feature and the stochastic trend structures, the widely utilised Gompertz model growth structure was inadequate in the quality of the in-sample fit compared to the more flexible models proposed. The only exception here were the countries with relatively small first wave epidemics which were Australia and Japan. These conclusions are quantitatively supported by the DIC results of Table 6.

We note that this is primarily due to the fact that the M2 model was not able to capture well the significant rate of change in the number of infections, and how this varied considerably over the calibration time period for most countries that experienced the most severe community spreading of COVID-19. Quite simply, models M2 and M4 were not sufficiently flexible to capture the structure of the first wave of the infection for the countries studied in general, and this would have implications on the performance of policy- and decision-making that is based around Gompertz growth curve models. This manifests as a clear model risk.

To diminish this risk, it is therefore recommended that policy-makers pay particular attention to the characteristics of the growth curves of the number of infections, and if necessary, step away from baseline to more advanced population growth models that accommodate the identified curve features.

### 7.2 Pre- and post-vaccination phases: January 2020—January 2021

This part of the study highlighted the importance of reviewing the decision for the best performing model as the pandemic evolves over time. We therefore see that in the case of the US we had to adapt the model and introduce an additional splice threshold to accommodate the rapid increase in the number of infections during the second wave of the epidemic, contrary to the rest of the countries for which the single-splice models were adequate.

We therefore recommend that decisions on the appropriate models be reviewed frequently while constantly incorporating newly available data the help understand the pandemic's evolution.

### 7.3 Out-of-sample forecast

The out-of-sample forecast results attest to the necessity of building more flexible models while taking into account the specificity of the infection growth rate in each country. Furthermore, the model risk induced and the repercussions if policymakers fail to do so is clearly demonstrated, as policy-making based on low-quality forecasts may have significant consequences both for the economy and the spread of the virus in the community. Should decisions

about response measures be taken based on the output of models that are not capable of correctly capturing the specificities of the growth curve, then this may have a tremendous cost on the community and the economy.

### 7.4 Stochastic observation models with news sentiment exposure-adjustment

As we had mentioned in Section 6.4, we make the premise that if a public health policy is being effectively communicated and people follow its guidelines, then the potential for community spread will be reduced over time, thereby resulting in reduced daily infection rates. Consequently, this study focused on the text-based sentiment exposure adjustment and we explored whether it could assist in model assessment and in reducing the risk of selecting an appropriate model.

As we saw, we indeed found that the sentiment exposure adjustment significantly improves the in-sample model fit, especially at the beginning of the pandemic. This is consistent with a perspective that the public were anxious in the period of January to April 2020 when a lot of uncertainty regarding the disease was present. The public were therefore much more receptive to the daily news announcements, as well as released public health warnings and the resultant policies and restrictions which were implemented to help reduce the potential for widespread community transmission. As the pandemic progressed there was a diminishing return on the model improvement through use of the sentiment exposure index. We believe that this is because people became more accustomed to the protection policies implemented, and the effect of news reporting on the society was, therefore, to a large extent saturated. Furthermore, much of the public policy statements were repeating and had already taken effect.

This analysis demonstrates the value of building such a component into a model to assess the effectiveness of news announcements, reporting approaches and policy decisions in a model-based framework. This could be used in future for aspects of scenario generation and assessment of policy communication approaches.

Even though this effect is clearly qualitatively verified in Figs 19–22, it is worth noting that it was not easy to discern via the DIC criterion which did not show significant improvement for the models with the sentiment adjustment. This is because the exposure change due to sentiment was compensated by the dispersion which got wider and therefore the likelihood surface and the DIC remained almost the same. However, once we look at the RMSE Table 9 the improvement becomes also quantitatively immediately evident. The big difference in the RMSE that we observe in the cases of the US and the UK with and without the sentiment adjustment, illustrates the improvement in the fit and it is clear that this is not due to the model choice. Consequently, the careful assessment of the fitted models with the most suitable diagnostic tools is paramount to further reduce model risk.

## 8 Conclusion

In this manuscript we have conducted a comparative study between different models for epidemic growth rate curves, which include the baseline popular Gompertz model and more flexible models which we are introducing into the literature of population models. Our goal was to demonstrate the risk associated with the selection of the appropriate population model when modelling the number of infected cases of an epidemic disease, and specifically the COVID-19 novel coronavirus.

We analysed seven countries with varying epidemic spread profiles (United Kingdom, Germany, Spain, Italy, United States, Japan, Australia) and we partitioned our analysis into the pre- and post-vaccination phases. We showed that the reference Gompertz model cannot

accommodate data that cover both phases, due to the specificities that the COVID-19 pandemic exhibited in the two periods under study, such as the rapid growth rate in the second wave of the pandemic starting in Autumn 2020. We interpret these results as a clear manifestation of the induced model risk, which may have significant repercussions if it is overlooked by policy-makers.

Furthermore, we constructed a novel sentiment index based on news articles and reporting about the COVID-19 pandemic from leading news sources (e.g. New York Times), institutions (e.g. WHO) and national Centres for Disease Control (US CDC, European CDC). We incorporated the sentiment index into our population growth models via an exposure adjustment, and we found that at the beginning of the pandemic the in-sample model fit is significantly improved if we include the sentiment index in the model. This is particularly important for model assessment and assessment of the effectiveness of the applied pandemic countermeasures and protective policies.

We believe that this work is an impactful contribution to the design and evaluation of countermeasures and their communication to people during extreme events, as well as scenario generation in preparation for addressing future crises. In the future, we aim to extend our sentiment index construction framework to multiple languages apart from English, which will allow us to use local news media per country for a more fine-grained analysis of the news reporting and its contribution in the modelling of the growth curve. In addition, we aim to partition the sentiment index into topics, e.g. health- or economic-related news sentiment, in order to better understand and evaluate the efficacy of the various policies and their impact. Finally, we will study different ways of incorporating the sentiment covariate in the population model in order to examine whether it can also enhance the out-of-sample predictive performance of the model, which would be critical for decision-makers.

## 9 Software

Code and data for reproducibility purposes are available at https://github.com/ichalkiad/covid19modelrisk.

## Supporting information

**S1 Appendix. Supplementary appendix containing algorithms used in the sentiment index construction, as well as analyses and results regarding Spain, Italy and Japan.**
(PDF)

**S1 Fig. In-sample fit results for Spain for the period January 2020—August 2020.** In-sample fitted plot (y-axis in log scale) for Spain by Model 2 (baseline, left) and Model 12 (best, right) (January 2020—August 2020).
(TIF)

**S2 Fig. In-sample fit results for Italy for the period January 2020—August 2020.** In-sample fitted plot (y-axis in log scale) for Italy by Model 2 (baseline, left) and Model 7 (best, right) (January 2020—August 2020).
(TIF)

**S3 Fig. In-sample fit results for Japan for the period January 2020—August 2020.** In-sample fitted plot (y-axis in log scale) for Japan by Model 2 (January 2020—August 2020).
(TIF)

**S4 Fig. In-sample fit results for Spain for the period January 2020—January 2021.** In-sample fitted plot (y-axis in log scale) for Spain by Model 2 (baseline, left) and Model 12 (best,

right) (January 2020—January 2021).
(TIF)

**S5 Fig. In-sample fit results for Italy for the period January 2020—January 2021.** In-sample fitted plot (y-axis in log scale) for Italy by Model 2 (baseline, left) and Model 12 (best, right) (January 2020—January 2021).
(TIF)

**S6 Fig. In-sample fit results for Japan for the period January 2020—January 2021.** In-sample fitted plot (y-axis in log scale) for Japan by Model 2 (baseline, left) and Model 12 (best, right) (January 2020—January 2021).
(TIF)

**S7 Fig. Out-of-sample forecast results for Spain.** Out-of-sample forecast plot for Spain by Model 2 (left) and Model 12 (right).
(TIF)

**S8 Fig. Out-of-sample forecast results for Italy.** Out-of-sample forecast plot for Spain by Model 2 (left) and Model 12 (right).
(TIF)

**S9 Fig. Out-of-sample forecast results for Japan.** Out-of-sample forecast plot for Spain by Model 2 (left) and Model 12 (right).
(TIF)

**S10 Fig. In-sample fit results with the sentiment exposure adjustment for Spain.** In-sample fitting plot for Spain by Model 2 for the first month with (left) and without (right) the sentiment exposure adjustment.
(TIF)

**S11 Fig. In-sample fit results with the sentiment exposure adjustment for Italy.** In-sample fitting plot for Italy by Model 2 for the first month with (left) and without (right) the sentiment exposure adjustment.
(TIF)

**S12 Fig. In-sample fit results with the sentiment exposure adjustment for Japan.** In-sample fitting plot for Japan by Model 2 for the first month with (left) and without (right) the sentiment exposure adjustment.
(TIF)

## Author Contributions

**Conceptualization:** Ioannis Chalkiadakis, Hongxuan Yan, Gareth W. Peters.

**Data curation:** Ioannis Chalkiadakis, Hongxuan Yan, Gareth W. Peters.

**Formal analysis:** Ioannis Chalkiadakis, Hongxuan Yan, Gareth W. Peters.

**Investigation:** Ioannis Chalkiadakis, Hongxuan Yan, Gareth W. Peters.

**Methodology:** Ioannis Chalkiadakis, Hongxuan Yan, Gareth W. Peters, Pavel V. Shevchenko.

**Project administration:** Gareth W. Peters.

**Resources:** Gareth W. Peters.

**Software:** Ioannis Chalkiadakis, Hongxuan Yan, Gareth W. Peters.

**Supervision:** Gareth W. Peters, Pavel V. Shevchenko.

**Validation:** Gareth W. Peters.

**Visualization:** Ioannis Chalkiadakis, Hongxuan Yan.

**Writing – original draft:** Ioannis Chalkiadakis, Hongxuan Yan, Gareth W. Peters.

**Writing – review & editing:** Ioannis Chalkiadakis, Hongxuan Yan, Gareth W. Peters, Pavel V. Shevchenko.

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
