## [Decision Letter · Decision Letter 0]

30 Apr 2021

PONE-D-21-11059

Infection rate models for COVID-19: model risk and public health news sentiment exposure adjustments

PLOS ONE

Dear Dr. Chalkiadakis,

Thank you for submitting your manuscript to PLOS ONE. After careful consideration, we feel that it has merit but does not fully meet PLOS ONE’s publication criteria as it currently stands. Therefore, we invite you to submit a revised version of the manuscript that addresses the points raised during the review process.

The manuscript has merit, but there are required several refinements and details towards sentiment index construction. Therewith, the English language should be improved.

We look forward to receiving your revised manuscript.

Kind regards,

Stefan Cristian Gherghina, PhD. Habil.

Academic Editor

PLOS ONE

Journal Requirements:

2) Please include captions for your Supporting Information files at the end of your manuscript, and update any in-text citations to match accordingly. Please see our Supporting Information guidelines for more information: http://journals.plos.org/plosone/s/supporting-information.

3) Please ensure that you refer to Figure 15 in your text as, if accepted, production will need this reference to link the reader to the figure.

Reviewers' comments:

Reviewer's Responses to Questions

**Comments to the Author**

1. Is the manuscript technically sound, and do the data support the conclusions?

Reviewer #1: Yes

Reviewer #2: Yes

Reviewer #3: Yes

2. Has the statistical analysis been performed appropriately and rigorously? 

Reviewer #1: Yes

Reviewer #2: Yes

Reviewer #3: Yes

3. Have the authors made all data underlying the findings in their manuscript fully available?

Reviewer #1: Yes

Reviewer #2: Yes

Reviewer #3: Yes

4. Is the manuscript presented in an intelligible fashion and written in standard English?

Reviewer #1: Yes

Reviewer #2: Yes

Reviewer #3: Yes

5. Review Comments to the Author

Reviewer #1: The authors present the model projection for COVID infection waves across seven countries. While the model approach has values, the following questions and issues need to be addressed.

1. Abstract: It does not reflect clearly the setting and context. The seven countries and three approaches/considerations must be mentioned to inform the reader adequately.

2. The countries selected and period selected should be clearly stated under a sub-heading like the reserach paper to appropriately inform the reader. At present it is lost in the text.

3. The source of the Language pricessing data - NYT has been considered. The NYT may be one of the richest source of news and public sentiment, it may not be adequate to reflect the public sentiment for other six countries. Why one leading source of news and public sentioment from the other countries were not considered? At least some comparative for the other countries and comparability between the news/public sentiments in NYT and other news sources should have been explored.

4. Discussion: Should be a separate segment and discuss the improvements/advantages and challenges compared to the real world setting observed and other models of prediction.

5. Conclusion: A section on Conclusion should be mentioned to emphasize the findings from the reserach and future implications.

6. Table 4, 5, 6: The fullform of DIC needs to be mentioned.

Reviewer #2: This paper models the temporal evolution of national level infection counts for a variety of countries by exploring a variety of stochastic population growth models. The authors characterize model risk mathematically, and develop an exposure adjustment to the force of infection comprised of a purposed built sentiment index constructed from a variety of authoritative public health reporting. They find that exposure adjustments that incorporate sentiment are better able to calibrate to early stages of infection spread in all countries under study.

The paper makes for a smooth read, while the methodology and empirical results are well described. The research is insightful for health policy making. My comments and suggestions are presented below.

1. GENERAL COMMENTS

Do the authors construct a single sentiment index based on the whole corpus collected from 5 different sources? This approach may bring some problems. For example, articles from New York Times make up the vast majority of the text sample. Although the paper argues that such a news source does attract a world-wide audience, at least in countries where English is amongst the official languages, I still think it is hard to be representative for other countries besides US. The reason is that the countermeasures taken by governments are mostly local based, thus people may not care about or be affected by anti-epidemic measures of other countries. Of course, the disease progress of other countries is also reported on New York Times, but people from other countries are most likely to pay attention to local newspapers.

I suggest the authors to construct a separate sentiment index for US using articles from New York Times and United States Centers for Disease Control and Prevention, and for European countries using European Centre for Disease Prevention and Control and United Nations Economic Commission for Europe. Alternatively, the authors could separate article contents in the corpus for different countries and construct corresponding sentiment index.

The results of this paper demonstrate that including a sentiment index significantly enhanced the model fit. I think this additional explanatory power of the news sentiment should be quantified to make the conclusion more convincing. Furthermore, does sentiment index improve out-sample forecasts in the study?

It is advised that the authors discuss the limitations of the study and give the future research work in conclusion part.

2. SPECIFIC COMMENTS

Some equations are too long and hard to read. Try to refine them using some notations.

I suggest the authors provide depict figure or descriptive statistics of the sentiment index.

There are some grammatical errors and/or instances of poorly worded/constructed sentences. Please carefully check the manuscript/abstract, and refine the language where necessary.

Reviewer #3: The authors have developed models to predict the infection rate of COVID-19 using public news sentiment. They find that exposure adjustments that incorporate sentiment are better able to calibrate to early stages of infection spread in all countries under study. It is interesting and of significance to practice. The authors stated that they used the sentiment analysis technique, however, they did not classify sentiment polarity. As it reads now, the sentiment analysis in this paper just sounds like topic or keyword analysis using dictionaries. Is this indeed the case? I suggest that the authors give more details about how they construct the dictionary.

6. PLOS authors have the option to publish the peer review history of their article (what does this mean?). If published, this will include your full peer review and any attached files.

Reviewer #1: No

Reviewer #2: **Yes: **Xuerong Li

Reviewer #3: No

---

## [Author Response · Author response to Decision Letter 0]

2 Jun 2021

Please find attached as a .pdf a comprehensive itemised list with the responses to each Reviewer comment included in the decision letter.

---

## [Decision Letter · Decision Letter 1]

4 Jun 2021

Infection rate models for COVID-19: model risk and public health news sentiment exposure adjustments

PONE-D-21-11059R1

Dear Dr. Chalkiadakis,

We’re pleased to inform you that your manuscript has been judged scientifically suitable for publication and will be formally accepted for publication once it meets all outstanding technical requirements.

Kind regards,

Stefan Cristian Gherghina, PhD. Habil.

Academic Editor

PLOS ONE

Additional Editor Comments (optional):

Reviewers' comments:

Reviewer's Responses to Questions

**Comments to the Author**

1. If the authors have adequately addressed your comments raised in a previous round of review and you feel that this manuscript is now acceptable for publication, you may indicate that here to bypass the “Comments to the Author” section, enter your conflict of interest statement in the “Confidential to Editor” section, and submit your "Accept" recommendation.

Reviewer #1: All comments have been addressed

Reviewer #2: All comments have been addressed

2. Is the manuscript technically sound, and do the data support the conclusions?

Reviewer #1: Yes

Reviewer #2: Yes

3. Has the statistical analysis been performed appropriately and rigorously? 

Reviewer #1: Yes

Reviewer #2: Yes

4. Have the authors made all data underlying the findings in their manuscript fully available?

Reviewer #1: Yes

Reviewer #2: Yes

5. Is the manuscript presented in an intelligible fashion and written in standard English?

Reviewer #1: Yes

Reviewer #2: Yes

6. Review Comments to the Author

Reviewer #1: The concerns and comments have been addressed by authors. The 'Limitations' section should be moved under the Discussion heading.

Reviewer #2: (No Response)

7. PLOS authors have the option to publish the peer review history of their article (what does this mean?). If published, this will include your full peer review and any attached files.

Reviewer #1: **Yes: **Manoja Kumar Das

Reviewer #2: **Yes: **Xuerong Li

---

## [Editor Report · Acceptance letter]

16 Jun 2021

PONE-D-21-11059R1 

Infection rate models for COVID-19: model risk and public health news sentiment exposure adjustments 

Dear Dr. Chalkiadakis:

I'm pleased to inform you that your manuscript has been deemed suitable for publication in PLOS ONE. Congratulations! Your manuscript is now with our production department. 

Kind regards, 

on behalf of

Dr. Stefan Cristian Gherghina 

Academic Editor

PLOS ONE